

# Separation of the optical and mass features of particle components in different aerosol mixtures by using POLIPHON retrievals in synergy with continuous polarized Micro-Pulse Lidar (P-MPL) measurements

Carmen Córdoba-Jabonero[1*], Michaël Sicard[2,3], Albert Ansmann[4], Ana del Águila[1], and Holger Baars[4]

[1]Instituto Nacional de Técnica Aeroespacial (INTA), Atmospheric Research and Instrumentation Branch, Torrejón de Ardoz (Madrid), Spain
[2]CommSensLab, Dept. of Signal Theory and Communications, Universitat Politècnica de Catalunya (UPC),
Barcelona, Spain
[3]Ciències i Tecnologies de l'Espai - Centre de Recerca de l'Aeronàutica i de l'Espai/Institut d'Estudis Espacials de Catalunya (CTE-CRAE/IEEC), Universitat Politècnica de Catalunya, Barcelona, Spain
[4]Leibniz Institute for Tropospheric Research (TROPOS), Leipzig, Germany

*Correspondence to*: Carmen Córdoba-Jabonero (cordobajc@inta.es)

**Abstract.** The application of the POLIPHON (POlarization-LIdar PHOtometer Networking) method in synergy with continuous 24/7 polarized Micro-Pulse Lidar (P-MPL) measurements to derive the vertical separation of two/three particle components in different aerosol mixtures, and the retrieval of their particular optical properties, is presented for the first time. The procedure of extinction-to-mass conversion, together with an analysis of the Mass Extinction Efficiency (MEE) parameter, is described, and the relative mass contribution of each aerosol component is also derived

in a further step. The general POLIPHON algorithm is based on the specific particle linear depolarization ratio given for different types of aerosols, and can be run in either 1-step (POL-1) or 2 steps (POL-2) versions in dependence on the either 2- or 3-component separation. In order to illustrate this procedure aerosol mixing cases observed over Barcelona (NE Spain) are selected: a dust event occurred on 5 July 2016; smoke plumes detected on 23 May 2016; and a pollination episode observed on 23 March 2016. In particular, the 3-component separation is just applied for the

dust case: a combined POL-1 with POL-2 procedure (POL-1/2) is used, and additionally the dust fine contribution to the total fine mode (dust fine plus non-dusty aerosols) is estimated. The high dust impact occurred in the first part of the day yields a mean mass loading of $0.6 \pm 0.1$ g m$^{-2}$ due to the prevalence of Saharan dust coarse particles in comparison with that obtained for the second part of the day, just a 34 % out of previous value, showing a rather weak dust incidence. In the smoke case, the arrival of fine biomass burning particles is detected at altitudes as high as 7 km

height. The smoke signature, also mixed with larger less depolarizing non-smoke aerosols, is observed along the day in dependence on the singular air masses origin with height, from either North America fires or the Arctic area, as reported by HYSPLIT backtrajectory analysis. The particle linear depolarization ratio for smoke shows values in the 0.10-0.15 range, even higher at given times, and the daily mean smoke mass loading is $0.017 \pm 0.008$ g m$^{-2}$, around 3 % out of that found for the dusty event. Pollen particles are detected up to 1.5 km height from 10:00 UTC on during

an intense pollination event with a particle linear depolarization ratio ranging between 0.10 and 0.15. The maximal mass loading of *Platanus* pollen particles is $0.011 \pm 0.003$ g m$^{-2}$, representing around 2 % out of the dust loading



during the higher dust incidence. Regarding the MEE derived for each aerosol component, their values are in agreement with other referenced in the literature for those specific aerosol types examined in this work: $0.5 \pm 0.1$ m$^2$ g$^{-1}$ and $1.7 \pm 0.2$ m$^2$ g$^{-1}$ are found for dust coarse and fine particles, respectively; $4.5 \pm 1.4$ m$^2$ g$^{-1}$ is derived for smoke, and $2.4 \pm 0.5$ m$^2$ g$^{-1}$ for non-smoke aerosols with Arctic origin (a MEE value close to that reported for Arctic aerosols: 2.17 m$^2$ g$^{-1}$, as supposed larger aerosols than those biomass burning particles); and a MEE of $2.4 \pm 0.8$ m$^2$ g$^{-1}$ is obtained for pollen particles, though it can reach higher/lower values depending on a predominant smaller/larger size of the pollen grains. Results reveal the high potential of the P-MPL system, a simple polarization-sensitive elastic backscatter lidar working in a 24/7 operation mode, to retrieve the relative optical and mass contributions of each aerosol component along all the day, reflecting the daily variability of their properties. Moreover, the method has the advantage to be relatively easily applicable also to spaceborne lidars with an equivalent configuration such as the ongoing Cloud-Aerosol Lidar with Orthogonal Polarization (CALIOP) onboard NASA/CALIPSO (Cloud-Aerosol Lidar and Infrared Pathfinder Satellite Observations), and the forthcoming Atmospheric Lidar (ATLID) onboard ESA/EarthCARE mission.

## 1 Introduction

It is widely known that atmospheric aerosols contribute to climate change due to their effects (direct and indirect) in the Earth's energy budget. Different types of aerosols present different radiative properties and thus contribute in a different way to climate change (Boucher et al., 2013; Myhre et al., 2013). As far as estimations of aerosol direct radiative forcing are concerned, the knowledge of the aerosol types under study is thus critical. The aerosol direct radiative properties involved in radiative transfer calculations are the particle extinction (scattering + absorption) coefficient, single scattering albedo (the ratio of scattering to extinction), asymmetry factor as defined as the intensity-weighted average cosine of the scattering angle, and their vertical distribution. Referring to the factors important in constraining the radiative effect of aerosols, Boucher et al. (2013) stated "Particularly important are the single scattering albedo (especially over land or above clouds) and the AOD", the aerosol optical depth, i.e. the column-integrated aerosol extinction. These two parameters can be estimated by or recalculated from the output of lidar-stand-alone algorithms such as Müller et al. (1999), Veselovskii et al. (2002) or Böckmann et al. (2005) which employ state-of-the-art elastic-Raman lidar measurements at several wavelengths. Such advanced measurements are scarce compared with the large database of elastic lidar measurements worldwide. For this reason, synergetic algorithms recently combine data from multi-wavelength elastic lidar and passive instrumentation to retrieve the extinction or both the extinction and the single scattering albedo at several wavelengths and discriminating between fine and coarse mode. Such algorithms are the LIdar-Radiometer Inversion Code-LIRIC (Chaikovsky et al., 2016), and the Generalized Aerosol Retrieval from Radiometer and LIDAR Combined data-GARRLiC (Lopatin et al., 2013). Now GARRLiC is embedded in a more generalized algorithm called the Generalized Retrieval of Atmosphere and Surface Properties inversion code-GRASP (Dubovik et al., 2014). The drawback of these algorithms is that they apply to at least three-wavelength elastic systems, while a majority of single- and dual-wavelength elastic systems are operating worldwide. For such systems, less sophisticated, the only way of discriminating between aerosol types is to have a



polarization-sensitive channel: the discrimination principle is based on the comparison of the particle depolarization ratio measured with two reference particle depolarization ratio values corresponding to two types of particles, one highly and one poorly depolarizing, previously identified. Such a method was first formulated by Chen et al. (2001) and then used by Shimizu et al. (2004) for the observation of Asian dust in China and Japan with one elastic and one depolarization sensitive channel. Since 2009 the method has been used in an increasing number of studies to discriminate between dust and smoke (Tesche et al., 2009; 2011); ash and fine mode particles (Ansmann et al., 2011; 2012; Sicard et al., 2012); pollen and background particles (Noh et al., 2013; Sicard et al., 2016a). Very recently this method, known as the POlarization-LIdar PHOtometer Networking (POLIPHON), has been refined by Mamouri and Ansmann (2014) to retrieve up to three aerosol components such as fine and coarse dust and non-dust particles. POLIPHON is also the basis of the retrieval of ice nuclei number concentration in desert dust layers (Mamouri and Ansmann, 2015) and cloud condensation nucleus number concentration (Mamouri and Ansmann, 2016).

In addition to their effects on climate, atmospheric aerosols are also known to have an important impact on human health when they are inhaled. For example, exposure to anthropogenic particles (pollution) is clearly identified as a public health hazard causing acute and chronic effects to the respiratory and cardiovascular systems (Dockery et al., 1993; Künzli et al., 2000; WHO, 2003). Airborne pollen grains produced by wind-pollinated plants are responsible of allergenic reactions when inhaled by humans (Cecchi, 2013). More recently Martiny and Chiapello (2013) highlighted the role of desert dust on meningitis epidemics. Toxicological studies are currently aiming to identify which particle characteristics are responsible for which adverse health effects (e.g., particle number, mass, size, surface, chemical composition). Among these properties what aerosol lidars can probably estimate the best is mass concentration when the aerosol type has been previously identified. However, mass concentration retrievals from lidar data are not common and there is very few information available on the vertical distribution of aerosol number and mass concentrations, although a number of field experiments involving research and commercial aircraft have measured aerosol concentrations (Heintzenberg et al.; 2011). Mass concentration profiles can be obtained by multiplying the lidar-derived extinction coefficient by the mass extinction efficiency, sometimes also called the specific extinction cross-section, when the latter is known or can be assumed. This conversion is often used to convert lidar-derived optical properties into mass concentration to test and evaluate transport models (Pérez et al., 2006; Sicard et al., 2015). Lately, POLIPHON is also used to extract from the total extinction the fractions of the high/moderate/low depolarizing particles which can then be converted separately into mass concentration (Mamouri and Ansmann, 2014; 2017). The method has been used for the estimation of the profile of mass concentration of dust (Ansmann et al., 2011; 2012), volcanic ash (Ansmann et al., 2012; Sicard et al., 2012) and pollen (Sicard et al., 2016b). It is worth mentioning that another field that would greatly benefit from the knowledge of the aerosol mass concentration profile is the air traffic, as large particles can damage aircraft engines. By way of example, let's recall the impact of the ash-loaded eruption plume from the Icelandic Eyjafjallajökull volcano on European air traffic in 2010 (Pappalardo et al., 2013).

The aim of this paper is to show the potential of simple lidar systems, with one elastic and one depolarization sensitive channel, to discriminate between several aerosol types and retrieve for each aerosol component the profiles of their optical properties and mass concentrations. The instrument used is the polarized version of the Micro-Pulse Lidar (P-MPL), the standard system within NASA/MPLNET (Micro Pulse Lidar NETwork) network (mplnet.gsfc.nasa.gov),





sited in the Universitat Politècnica de Catalunya (UPC) in Barcelona (BCN) at the northeastern Spain. The P-MPL is
an elastic and monochromatic low-energy system which includes also a depolarization-sensitive channel, operating in
an automatic and continuous 24/7 mode. The algorithm used to optically discriminate components in aerosol mixtures
is the POLIPHON method, both 1-step and 2-step versions, in order to assess the vertical separation of a maximum of
three aerosol components. The synergetic use of P-MPL/POLIPHON is tested with aerosol mixtures containing
specific climate-relevant aerosols, namely desert dust, fire smoke and pollen. It should be noted that this is the first
time that POLIPHON, well established for sophisticated powerful European Aerosol Research Lidar NETwork
(EARLINET, www.earlinet.org) lidars, is applied to worldwide and continuous simple elastic P-MPL measurements.
Moreover, the method has the advantage to be relatively easily applicable also to spaceborne lidars with an equivalent
configuration such as the ongoing Cloud-Aerosol Lidar with Orthogonal Polarization (CALIOP) onboard
NASA/CALIPSO (Cloud-Aerosol Lidar and Infrared Pathfinder Satellite Observations) which has two elastic and one
depolarization-sensitive channel, and the forthcoming Atmospheric Lidar (ATLID) onboard EarthCARE (future ESA
mission to be launched in 2019) which will have a high-spectral resolution receiver and a depolarisation channel.

The paper is organized as follows: **Section 1** presents the introductory framework; the methodology is introduced in
**Section 2**, which breaks down in the description of the measurement station and of the selected aerosol cases (**Sect.
2.1**), as well as the lidar system used in this paper (**Sect. 2.2**), an extended overview of the POLIPHON method (**Sect.
2.3**) and a detailed extinction-to-mass conversion procedure (**Sect. 2.4**); **Section 3** shows the results and their
discussion for each case (dust, smoke and pollen). Finally, a summary of the work and the main conclusions are
presented in **Section 4**. In addition, a list of acronyms (symbols) identifying the parameters/variables used in the work
is shown in **Appendix A**.

## 2 Methodology

### 2.1 Measurement station and selected aerosol case studies

Barcelona (BCN) station is an urban site located at the North East Iberian Peninsula (41.4ºN, 2.1ºE, 115 m a.s.l.), by
the coast of the Mediterranean Sea, in the North campus of the Universitat Politècnica de Catalunya (UPC) at the
centre of the Barcelona city. The typical background aerosol is a mixing of polluted particles with a minor contribution
of marine aerosols, only predominant under particular clean conditions; other aerosol types, such as desert dust, fire
smoke, pollen, etc., are also frequently found (Sicard et al., 2011). BCN is a well-established EARLINET station
besides a recent MPLNET site, where a polarized Micro-Pulse Lidar (P-MPL) is in routine operation since 2014. BCN
is also a NASA/AERONET (Aerosol Robotic NETwork, aeronet.gsfc.nasa.gov) site.

In this work, three case studies of different aerosol mixtures (dust, fire smoke and pollen, all mixed with local
background aerosols) observed over BCN are examined in order to introduce the combined application of POLIPHON
in synergy with continuous P-MPL measurements for the separation of, in particular, Saharan dust aerosols, fire smoke
plumes and pollen particles from other aerosols mixed with them. Those selected dust, smoke and pollen cases
occurred on 5 July, 23 May and 23 March 2016, respectively. HYSPLIT backtrajectory (Hybrid Single Particle
Lagrangian Integrated Trajectory model Version 4 developed by the NOAA's Air Resources Laboratory (ARL);





Draxler and Hess, 1998; Stein et al., 2015, Rolph et al., 2017) analysis is used to confirm the presence of dust and
smoke over BCN for each particular case. HYSPLIT backtrajectories are calculated for those days ending over BCN
at given altitudes and several times in relation with the results obtained and discussed later in **Section 3** for the dust
and smoke cases. In particular, the 5-day backtrajectory analysis indicates Saharan air masses arriving at high altitudes
(> 2000 m a.g.l.) on 5 July 2016 only for the first part of the day, meanwhile North Atlantic air masses are arriving at
lower heights (see **Fig.1**, a-c panels); during the second part of the day, air masses at any altitude are also mostly
coming from North Atlantic and central Spain regions (see **Fig. 1**, d-f panels), but not from Saharan desert. On the
other hand, smoke plumes detected on 23 May 2016 over BCN seem to be arriving from North America fires using
10-day backtrajectories; depending on the altitude and time of the arrival, air masses are coming from either Canada
and USA areas carrying fine biomass burning particles or Artic region with larger aerosols in comparison with those
smoke particles (see **Fig. 1**, g-l panels). The pollen case was selected in the period March-April as the day with the
highest peak of daily pollen concentration. Such a peak occurred on 23 March 2016 and the most abundant taxon was
*Platanus*. Belmonte (2016) counted a near-surface concentration of around 1700 grains of *Platanus* taxon per cubic
meter in Barcelona downtown on 23 March 2016. This value is close to the daily values found in the pollination event
of March 2015 also in Barcelona described by Sicard et al. (2016) as particularly strong in terms of pollen
concentration. These results will be discussed in detail together with those obtained for each aerosol case in **Section
3**.

**2.2 Polarized Micro-Pulse lidar (P-MPL) system**

The polarized Micro-Pulse lidar system (P-MPL v. 4B, Sigma Space Corp.) acquires vertical aerosol profiles with a
relatively high frequency (2500 Hz) using a low-energy (~ 7 µJ) Nd:YLF laser at 532 nm. The P-MPL acquisition
settings follow the NASA/MPLNET requirements of 30 s integrating time and 15 m vertical resolution. Polarization
capabilities rely on the collection of two-channel measurements, i.e., the signal measured in the so-called 'co-polar'
and 'cross-polar' channels of the instrument, denoted as $ch_{co}(z)$ and $ch_{cr}(z)$, respectively (see Sigma Space Corp.
Manual, 2012, for more details).

$\delta^V$ is defined as (hereafter, the dependence with height is omitted for simplicity)

$$\delta^V = \frac{P^s}{P^p},$$    (1)

where $P^p$ and $P^s$ represent, respectively, the parallel and perpendicular P-MPL range-corrected signals (RCS, also
called Normalized-Relative-Backscatter signals, $NRB$). By adapting the methodology described in Flynn et al. (2007),
the linear volume depolarization ratio $\delta^V$ for a MPL system can be easily expressed as

$$\delta^V = \frac{ch_{cr}}{ch_{co}+ch_{cr}}.$$    (2)

Indeed, both RCS signals can be expressed in terms of those P-MPL co- and cross-channels, i.e., $P^p = ch_{co} + ch_{cr}$
and $P^s = ch_{cr}$ (see Flynn et al., 2007, for more details), being the total RCS: $P^{tot} = P^p + P^s = ch_{co} + 2\,ch_{cr}$. Final
corrected $P^{tot}$, $P^p$ and $P^s$ are obtained using the procedure described in Campbell et al. (2002) and Welton and
Campbell (2002). In order to increase the signal-to noise ratio (SNR), both $P^p$ and $P^s$ are hourly-averaged signals in
this work. However, higher uncertainties are found for daytime measurements due to the SNR decrease. Relative





uncertainties estimated for the main parameters as derived from P-MPL measurements are shown in **Table 1**
(references included).

The particle linear depolarization ratio $\delta_p$ is calculated by the procedure shown in Cairo et al. (1999), and expressed as

$$\delta_p = \frac{R \times \delta^V \times (\delta_{mol}+1) - \delta_{mol} \times (\delta^V+1)}{R \times (\delta_{mol}+1) - (\delta^V+1)}, \tag{3}$$

where $R$ is the backscattering ratio ($R = \frac{\beta_m + \beta_p}{\beta_m}$), being $\beta_m$ and $\beta_p$ the molecular and particle backscatter coefficients,
respectively; and $\delta_{mol}$ is the molecular depolarization ratio. In particular, the filters of the P-MPL optical receiving system presents a spectral band lower than 0.2 nm (Sigma Space Corp. Manual, 2012), producing a temperature-independent $\delta_{mol}$ of 0.00363 according to Behrendt and Nakamura (2002). The particle backscatter coefficient $\beta_p$ is obtained by applying the Klett-Fernald (KF) algorithm (Fernald, 1984; Klett, 1985) to $P^{tot}$ ($= P^p + P^s$) profiles obtained from P-MPL measurements in synergy with simultaneous sun-photometer measurements that provide
ancillary data of the Aerosol Optical Depth (AOD), that is the constraint condition for KF inversion convergence. Hence, a vertically-averaged lidar ratio (LR, extinction-to-backscatter ratio, denoted as $S_a$) can be also estimated by using this KF iterative approach in P-MPL measurements, since the LR value varies in each iteration, reaching the convergence once the relative difference between the lidar-derived height-integrated particle extinction profile $\tau^{MPL}$ ($= \sum_z \sigma_p(z) = \sum_z [S_a \times \beta_p(z)]$) and the AERONET AOD is lower than a given convergence factor (see Córdoba-
Jabonero et al., 2014, for more details of this iterative convergence method applied to specific MPL measurements). In this study, a convergence factor of 1 % is applied (relative uncertainties found for $S_a$ are 5-10 %, see **Table 1**). AERONET data, both AOD and the Ångström exponent (AEx), are also hourly-averaged in order to coincide with the 1-h averaging applied to P-MPL measurements.

**2.3 POLIPHON method**

**2.3.1 General features**

The POLIPHON (POlarization-LIdar PHOtometer Networking) method was developed at the Leibniz Institute for Tropospheric Research (TROPOS, www.tropos.de) for application in polarization-lidar measurements in order to separate the optical properties (backscatter, extinction) of aerosol mixtures into their components with clearly different particle depolarization ratios. POLIPHON can run two ways: as 1-step retrieval (POL-1 approach hereafter) or in 2
steps (POL-2 approach hereafter), retrieving the separation of two or three aerosol components, respectively. A complete description of the POLIPHON discrimination technique can be found in Mamouri and Ansmann (2014). In particular, the POL-1 approach is successfully applied for separation of dust from biomass burning smoke particles (Tesche et al., 2011; Ansmann et al., 2012), and volcanic ash aerosols from other fine particles (Ansmann et al., 2012); and the POL-2 approach is used for partition of dust coarse and fine components and their discrimination from other
non-dusty aerosols (marine, anthropogenic pollution) (Mamouri and Ansmann, 2017).

In this work, as stated before, the separation of the optical properties of dust, smoke and pollen particles from their mixtures with other aerosols is performed by applying POLIPHON to P-MPL measurements. The POL-1 approach



(2-component separation) is used for the selected smoke and pollen cases as occurred on 23 May 2016 and 23 March 2016, respectively, over BCN, in order to discriminate the smoke (SM) signature from other non-smoke (NS) aerosols,

and the pollen (PL) particles from other local background aerosols (BA). The dust case observed on 5 July 2016 is examined to present the separation into three components: dust coarse (Dc), dust fine (Df) and non-dusty (ND) aerosols. However, particularly for this case, instead of the POL-2 approach only, a combined version of POLIPHON using together both POL-1 and POL-2 approaches (namely POL-1/2) is applied (Mamouri and Ansmann, 2017). A more detailed description of this POL-1/2 retrieval, and its use in this work, is shown in the next **Section 2.3.2**. In

general, one of the constraints of POLIPHON is that it is based on the appropriate selection of the linear depolarization ratio for each 'pure' (not mixed) type of specific aerosols. **Table 2** shows the particular $\delta_i$ values assumed for each specific ($i$) aerosol component. In particular, in the dust case $i = 1$ is denoted for total dust (DD), and 2 for non-dust (ND) by using POL-1, and $i = 1$ for dust coarse (Dc), 2 for dust fine (Df), and 3 for non-dust (ND) by using POL-2; in the smoke case $i = 1$ stands for smoke (SM), and 2 for non-smoke (NS) by using POL-1; and in the pollen case $i =$

1 is for pollen (PL), and 2 for local background aerosols (BA), likely a mixture of small pollution particles mostly present in an urban environment as Barcelona city, by using POL-1. After separation of the different aerosol components, the respective extinction coefficients are calculated by assuming LR values typical for each aerosol type: 55 sr for dust (Dc and Df components) (Mamouri and Ansmann, 2014), 70 sr for smoke plumes (Groβ et al., 2013), and 50 sr for pollen particles (Sicard et al., 2016).

Moreover, the backscatter fraction for each aerosol component is presented along the day, as expressed in terms of the relative ratio between the specific height-integrated backscatter coefficient for each aerosol component, $\overline{\beta_i}$, and the total (sum of all the components) height-integrated particle backscatter coefficient, $\overline{\beta_p}$, i.e., the $\frac{\overline{\beta_i}}{\overline{\beta_p}}$ ratio (%), as calculated from the continuous 24/7 P-MPL measurements.

**2.3.2 POL1/2 approach applied to the dust case: combined POL-1 and POL-2 versions**

In dusty events, POL-1 is used to separate dusty (DD) from non-dusty (ND) aerosols; instead, POL-2 is a 2-step approach used to first (step 1) separate Dc particles from the total fine mode (Df + ND) (ND are assumed to be only fine aerosols as composed mostly of small pollution particles, since AODs are large enough for neglecting the marine impact), and then (step 2) that fine contribution is separated into Df and ND particles (see more details in Mamouri and Ansmann, 2014). In the overall POL-2 procedure, the depolarization ratio for the total fine (Df+ND) mixture (i.e.,

the residual fine depolarization ratio), $\delta_{Df+ND}$, must be either assumed or known. In our case, $\delta_{Df+ND}$ can be estimated by a combined algorithm that uses both POL-1 and POL-2 versions (POL-1/2), as also reported by Mamouri and Ansmann (2017). In particular, the statement that the backscatter coefficient profiles obtained from the POL-1 retrieval for the DD (Dc+Df) component, $\beta_{DD}(z)|_{POL-1}$, is identical to the sum of the backscatter coefficient profiles for the dust coarse (Dc) and dust fine (Df) retrieved independently by the POL-2 version (i.e., $\beta_{Dc}(z)|_{POL-2}$ and

$\beta_{Df}(z)|_{POL-2}$, respectively) must be fulfilled; that is,

$$\beta_{DD}(z)|_{POL-1} = \beta_{Dc}(z)|_{POL-2} + \beta_{Df}(z)|_{POL-2}. \tag{4}$$


For that purpose, first, $\beta_{DD}(z)|_{POL-1}$ profiles are derived; then, a set of both $\beta_{Dc}(z)|_{POL-}$ and $\beta_{Dc}(z)|_{POL-2}$ is obtained for several $\delta_{Df+ND}$ values ranging between the specific depolarization ratios of Df particles ($\delta_{Df}$=0.16) and ND aerosols ($\delta_{ND}$=0.05) (see **Table 2**). Those $\delta_{Df+ND}$ are iteratively introduced with steps of 0.01 in the POL-2 approach point-to-point along the whole profile in order to obtain an optimal $\delta_{Df+ND}(z)$ profile, which must satisfy that the two terms of the equality in **Eq. (4)** are equal at each $z$-point. For instance, the minimal value obtained for the root square differences, $\Delta$, between both terms in **Eq. (4)** at a given $z$, i.e.,

$$\min\{\Delta(z)\} = min\left\{\sqrt{\left[\beta_{DD}(z)|_{POL-1} - \left(\beta_{Dc}(z)|_{POL-2} + \beta_{Df}(z)|_{POL-2}\right)\right]^2}\right\} \tag{5}$$

is used as proxy in that iteration process. Hence, once those $min\{\Delta\}$ are achieved for a given $\delta_{Df+ND}$ along the whole profile, the optimal vertical $\delta_{Df+ND}(z)$ profile is determined. Moreover, since $\delta_{Df+ND}(z)$ is defined in a good approximation as

$$\delta_{Df+ND}(z) = \delta_{Df} \times \gamma(z) + \delta_{ND} \times (1 - \gamma(z)) \tag{6}$$

where $\gamma(z)$ and $(1-\gamma(z))$ are, respectively, the fraction of each Df and ND components as contributed to the total fine (Df+ND) mode mixture, this contribution of each aerosol fine component to the total fine mode can also be estimated with height, i.e., $\gamma(z)$ is thus determined.

Once the profile of $\delta_{Df+ND}$ (and $\gamma$) is optimally determined , the total particle backscatter coefficient profiles $\beta(z)$ can be separated into all three components ($\beta_{Dc}$, $\beta_{Df}$ and $\beta_{ND}$) for the dust case by applying POL-2 (step 2) retrieval (see Mamouri and Ansmann, 2014, for more details). Hence, their relative contribution (i.e., the $\frac{\overline{\beta_i}}{\overline{\beta_p}}$ ratio, %) can be also derived.

For comparison, a columnar $\delta_{Df+N}^c$ value is also calculated using the same POLIPHON procedure as described before, but the minimum of the root mean square differences, $\tilde{\Delta}$, between both terms in **Eq. (4)**, i.e.,

$$\min\{\tilde{\Delta}\} = min\left\{\sqrt{\frac{\left[\Sigma_z\left[\beta_{DD}(z)|_{POL-1} - (\beta_{Dc}(z)|_{POL-2} + \beta_{Df}(z)|_{POL-2})\right]^2\right]}{n}}\right\} \tag{7}$$

is used instead as the proxy applied in the iterative retrieval ($n$ stand for the number of z-points along the overall profile). For instance, **Figure 2** shows the particle backscatter coefficients profiles as obtained from either POL-1 ($\beta_{DD}$ and $\beta_{ND}$) or POL-1/2 ($\beta_{Dc}$ and $\beta_{Df}$, being $\beta_{Dc} + \beta_{Df} = \beta_{DD}$, and $\beta_{ND}$) approaches at two times (02:00 and 16:00 UTC) on 5 July 2016, using both the optimal $\delta_{Df+ND}(z)$ profile (**Fig. 2a**), and the columnar $\delta_{Df+ND}^c$ (**Fig. 2b**). Discrepancies are observed in both the dust and non-dust components by using a single columnar $\delta_{Df+ND}^c$ value instead of the optimal $\delta_{Df+ND}(z)$ profile. For comparison between **Fig. 2a** and **2b**, differences are clearly found in $\beta_{ND}$ at 02:00 UTC, picked at around 4.5 km height, as derived from either POL-1 or POL-1/2, in addition to those found for $\beta_{DD}$ in comparison with $\beta_{Dc}$ and $\beta_{Df}$ (particularly evident at 16:00 UTC, with $\beta_{DD} \ll \beta_{Df}$ between 1 and 2 km height) (see **Fig. 2b**). These results highlight the use of a height-resolved $\delta_{Df+N}$ rather improves the retrieval. Indeed, the use of a single columnar (no height-resolved) $\delta_{Df+ND}^c$ (and $\gamma^c$) in the retrieval can be inadequate due to the plausible variability of the relative fraction of Df particles to the total fine (Df+ND) mode with height. In particular, this is





corroborated looking at the optimal height-averaged $\overline{\delta_{Df+N}}$ values obtained at 02:00 and 16:00 UTC are, respectively:

$0.12 \pm 0.04$ ($\bar{\gamma} = 66 \pm 32$ %) and $0.09 \pm 0.05$ ($\bar{\gamma} = 40 \pm 38$ %), in comparison with those columnar $\delta_{Df+N}^{c}$ values found

at 02:00 and 16:00 UTC, respectively: 0.14 ($\gamma^{c} = 82$ %) and 0.06 ($\gamma^{c} = 9$ %).

**2.4 Extinction-to-mass concentration conversion**

**2.4.1 General procedure**

The conversion from extinction ($\sigma$, m$^{-1}$) to mass concentration ($M$, g m$^{-3}$) is performed for each component ($i$) by

means of the so-called Mass Extinction Efficiency (MEE, or also mass-specific extinction coefficient) ($k$, m$^2$ g$^{-1}$) by

using the following relationships (Ansmann et al., 2012; Córdoba-Jabonero et al., 2016) at each altitude $z$:

$$M_i(z) = \frac{\sigma_i(z)}{k_i}. \tag{8}$$

The effective MEE ($k_{eff}$, m$^2$ g$^{-1}$), linking the total aerosol extinction from all aerosol components (i.e., AOD) to the

Total Mass Concentration (TMC), is given by:

$$k_{eff} = \frac{AOD}{TMC}, \tag{9}$$

where $TMC = \sum_i \overline{M_i}$ represents the total mass loading in g m$^{-2}$, with $\overline{M_i}$ the height-integrated mass concentration for

each component (i.e., $\overline{M_i} = \sum_z M_i(z) \, \Delta z$, with $\Delta z$ the height resolution). $k_{eff}$ is a measure of the predominant particle

size; $k_{eff}$ values lower and higher than 1.5 m$^2$ g$^{-1}$ are representative of large and small particles, respectively, as

reported by the Optical Properties of Aerosols and Clouds database (OPAC; www.pole-ether.fr). The mass

contribution or fraction of each aerosol component is expressed by the relative ratio between $\overline{M_i}$ and $TMC$, i.e.,

$\overline{M_i}/TMC$ (%).

Columnar MEE values can be obtained from AERONET data and the particle density ($Pd$, g cm$^{-3}$) assumed for each

aerosol component examined in this work by using the expression (Ansmann et al., 2012):

$$k_{c,f} = \frac{\tau_{c,f}}{Pd \times VC_{c,f}} = \frac{1}{Pd \times c_{v_{c,f}}}, \tag{10}$$

where $k_{c,f}$ designate the MEE for coarse and fine modes, as denoted by subscripts 'c' and 'f', respectively; similarly,

$VC_{c,f}$ (10$^{-12}$ Mm) and $\tau_{c,f}$ are the AERONET volume concentrations and extinction values, respectively, for the coarse

and fine modes. $c_{v_{c,f}}$ ($= \frac{VC_{c,f}}{\tau_{c,f}}$) are the corresponding so-called extinction-to-volume conversion factors.

Indeed, our strategy is to obtain the actual $c_{v_{c,f}}$ values, and then the $k_{c,f}$ using typical particle densities, from

AERONET sun-sky photometer observations performed simultaneously with P-MPL observations, as long as the

separated aerosol components can be identified as composed of pure coarse or fine particles. **Table 3** shows the

AERONET parameters involved in the extinction-to-mass conversion ($VC_{c,f}$, $\tau_{c,f}$) at selected times for each aerosol

case together with those typical particle densities $Pd$ for each aerosol component. In particular, $Pd$ values assumed

for each type of aerosols are: 2.60 g cm$^{-3}$ for dust (Ansmann et al., 2012), 1.30 g cm$^{-3}$ for smoke (Reid et al., 2005),

0.92 g cm$^{-3}$ for pollen (*Platanus*) particles (Jackson and Lyford, 1999; Zhang et al., 2014). For the other components,

the particle density is obtained from the OPAC database (Hess et al., 1998): a particle density $Pd = 1.8$ g cm$^{-3}$ is




assumed for both the ND and BA components in the dust and pollen cases, respectively, corresponding to background urban aerosols, mostly composed of fine pollution particles; and for the NS component in the smoke case a $Pd_{NS} =$ 2.0 g cm$^{-3}$, as reported by OPAC for Arctic aerosols, is assumed since the NS signature is found when air masses are coming from the Arctic as indicated by backtrajectory analysis (see **Sect. 2.1**). However, the corresponding $c_v$ and $k$

values must be examined in more detail in the extinction-to-mass conversion procedure for each aerosol case, as explained next.

**2.4.2 Dust case**

As stated before, POL-1/2 retrieval is applied to separate three components for the dust case ($i$ = Dc, Df and ND). Conversion factors are only reported for coarse and fine mode particles in overall using AERONET data (**Eq. 10**). In

this case, the coarse mode is completely composed by Dc particles (the ND component is assumed to be fine aerosols only, see **Sect. 2.3**). Hence, the MEE for Dc particles, $k_{Dc}$, is easily obtained from

$$k_{Dc} = \frac{\tau_c}{Pd_{Dc} \times VC_c} = \frac{1}{Pd_{Dc} \times c_{v_c}} \qquad (11)$$

with $Pd_{Dc}$ = 2.6 g cm$^{-3}$ for dust. However, MEE for Df particles, $k_{Df}$, and ND aerosols, $k_{ND}$, must be determined from the MEE value obtained for the total fine (Df+ND) mode, $k_{Df+ND}$, that is,

$$k_{Df+ND} = \frac{\tau_f}{Pd_{Df+ND} \times VC_f} = \frac{1}{Pd_{Df+ND} \times c_{v_f}}, \qquad (12)$$

where $Pd_{Df+ND}$ represents a weighted value of the particle density for the overall fine (Df+ND) mode. Once estimated $\delta_{Df+ND}$, and $\gamma$ (see **Eq. 6**), $Pd_{Df+N}$ can be expressed as

$$Pd_{Df+ND} = Pd_{Df} \times \gamma + Pd_{ND} \times (1 - \gamma), \qquad (13)$$

where $Pd_{Df}$ and $Pd_{ND}$ are the particle densities assumed for dust (2.6 g cm$^{-3}$) and non-dust aerosols (1.8 g cm$^{-3}$),

respectively (**Table 3**). Hence, the height-integrated mass concentration for the total fine (Df+ND) mode, $\overline{M_{Df+ND}}$, can be calculated from

$$\overline{M_{Df+ND}} = k_{Df+N}^{-1} \times \tau_{Df+ND} = \overline{M_{Df}} + \overline{M_{ND}}, \qquad (14)$$

where $k_{Df+ND}$ is calculated from **Eq. 12**, and $\overline{M_{Df}}$ and $\overline{M_{ND}}$ are, respectively, the mass concentrations for Df and ND aerosols (note that these quantities are height-integrated variables, i.e., mass loadings). In particular, $\overline{M_{Df}}$ can be

determined by assuming a representative conversion factor $c_v$ for Df particles, since

$$\overline{M_{Df}} = \tau_{Df} \times Pd_{Df} \times c_{v_{Df}}. \qquad (15)$$

Mamouri and Ansmann (2017) reported statistical AERONET-based extinction-to-mass conversion factors for dust fine particles $c_{v_{Df}}$ in the interval of 0.21-0.25 ($\pm$ 0.05) 10$^{-12}$ Mm. In this work, this set of values is introduced in the algorithm in order to obtain an optimal $c_{v_{Df}}$ value satisfying the following condition: $\overline{M_{Df}} < \overline{M_{Df+ND}}$, being estimated

$\overline{M_{Df}}$ from **Eq. 15**. At the same time, $\overline{M_{ND}}$ is also obtained, since

$$\overline{M_{ND}} = \overline{M_{Df+ND}} - \overline{M_{Df}}. \qquad (16)$$

Hence, $k_{Df}$ and $k_{ND}$ (and $c_{v_{ND}}$) are calculated applying, similarly to **Eqs. 10-12**, the following expressions:





$$k_{Df} = \frac{1}{Pd_{Df} \times c_{vDf}},$$ (17)

$$k_{ND} = \frac{\tau_{ND}}{\overline{M_{ND}}},$$ (18)

and

$$c_{vND} = \frac{1}{Pd_{ND} \times k_{ND}}.$$ (19)

Otherwise, $\overline{M_{Df}} = \overline{M_{Df+ND}}$ (and then, $k_{Df} = k_{Df+ND}$), and $\overline{M_{ND}} = 0$. Finally, the total mass concentration $TMC$ (i.e., mass loading, in g m$^{-2}$) is obtained from

$$TMC = \overline{M_{Dc}} + \overline{M_{Df+ND}} = \overline{M_{Dc}} + \overline{M_{Df}} + \overline{M_{ND}}.$$ (20)

Those AERONET parameters used in the extinction-to-mass conversion together to the particular $c_v$ and $k$ values obtained at some explicit times (see **Table 3**) are in agreement with those reported by other authors (i.e., Mamouri and Ansmann, 2014; 2017) for dust. In addition, $k_{ND}$ values are derived between 2.52 and 2.92 m$^2$ g$^{-1}$, similar to those reported by OPAC for urban aerosols (2.87 m$^2$ g$^{-1}$), as assumed for the ND component in this work.

### 2.4.3 Smoke and pollen cases

For both these cases, optical properties are separated into two aerosol components by using POL-1 approach. Hence, mass concentrations are derived directly from **Eqs. 8-10** of the general extinction-to-mass conversion procedure using AERONET data, satisfying that each component is composed mostly of either coarse or fine mode particles, as described in **Section 2.4.1**.

In particular, the smoke (SM) component is supposed the fine mode as composed of fine biomass burning particles, and the coarse mode is associated to the non-smoke (NS) component by assuming particles larger than smoke coming from the Arctic area. For instance, a $k_{SM} = 4.5 \pm 1.4$ m$^2$ g$^{-1}$ is derived for fine smoke particles at 06:00 UTC (see **Table 3**); this value is in good agreement with that reported for Canadian forest fire smoke aerosols by other authors (Ichoku and Kaufman, 2005; Reid et al., 2005). However, a rather lower MEE value is obtained for the coarse mode NS particles ($k_{NS} = 2.4 \pm 0.5$ m$^2$ g$^{-1}$) at the same time. In the pollen case, PL particles are predominantly large particles in comparison with the fine (and less depolarizing) component corresponding to local background aerosols (BA), as assumed composed of small polluted particles of urban origin (marine contribution is neglected, as stated in **Sect. 2.**). For instance, a $k_{PL} = 2.3 \pm 0.1$ m$^2$ g$^{-1}$ is obtained for pollen particles at 15:00 UTC, when pollination event is enhanced, as described later in **Section 3.3**.

**Table 3** shows the derived MEE values ($k$, m$^2$ g$^{-1}$) at selected times by using the corresponding $c_v$ factors and the assumed particle densities ($Pd$, g cm$^{-3}$) for each component. Particular similarities and discrepancies found from those assumptions will be discussed in more detail in **Section 3**.

### 3 Results

### 3.1 Dust case



A dusty event occurred over BCN station on 5 July 2016, mostly intense during the first part of the day as also

confirmed by AERONET data with moderate AOD and AEx < 0.5 values together with HYSPLIT backtrajectory

analysis (**Sect. 2.1**). The separation into three components (Dc, Df and ND) of dusty mixtures using the synergy of

hourly-averaged P-MPL measurements and POL-1/2 retrieval is performed along the day. Prior using POL-1/2,

vertical profiles of the total particle backscatter coefficient ($\beta_p$), as derived from the KF algorithm (if the KF retrieval

is feasible, estimated LR values are discussed later), and the linear particle depolarization ratio ($\delta_p$) are obtained along

the day. Then, the corresponding vertical profiles of the backscatter coefficients for each specific component ($\beta_i$, $i =$

Dc, Df, ND) are retrieved by using POL-1/2 (**Sect. 2.3.2**). The three specific depolarization ratios selected for each

pure aerosol component ($\delta_i$, $i =$ Dc, Df, ND), required for the POL-1/2 retrieval, are shown in **Table 2**. As mentioned

before, height-integrated values of all these backscatter coefficient profiles ($\overline{\beta_p}$, and the three $\overline{\beta_i}$ for each component)

are calculated along the 24 hours of the day (if the KF retrieval is feasible) to obtain the daily temporal evolution of

the optical contribution for each aerosol component in terms of their specific relative ratio $\frac{\overline{\beta_i}}{\overline{\beta_p}}$ (in %). Regarding the

height-integrated mass concentration ($\overline{M_i}$, $i =$ Dc, Df, ND; **Sect. 2.4**), the daily evolution of specific mass contribution

ratio, i.e., the relative ratio $\frac{\overline{M_i}}{TMC}$ (in %), is also calculated for each aerosol component (note that height-integrated mass

concentrations represent the mass loading, expressed in g m$^{-2}$). For simplicity, the same notation is used for mass

concentration and mass loading.

**Figure 3** shows the daily evolution of the specific (a) optical and (b) mass relative contribution for each aerosol

component along the day. A high loading of large particles with peaks of 78 % for $\beta_{Dc}$ and 98 % for $M_{Dc}$ is obtained

in the first half of the day. These peaks drop to minimums of 9 and 43 %, respectively, in the second part of the day.

In this period of the day, the optical contribution of the total dust (Dc+Df) varies between 17 and 46 % while the mass

contribution ratio varies between 56 and 98 %. In terms of mean $TMC$ (dust loading), values of 0.6 ± 0.1 and 0.2 ±

0.1 g m$^{-2}$ are estimated, respectively, at time intervals before and after noon: the last one just represents a $TMC$ of 34

% respect to that found for the first part of the day. Specific $\overline{M_i}$ and $TMC$ at given times are shown in **Table 4**.

Therefore, two different dusty scenarios with an intense and weak dust impact are clearly observed in the first and

second part of the day, respectively.

These results are related to the mean MEE values found for dust particles: $k_{Dc} = 0.5 ± 0.1$ m$^2$ g$^{-1}$ and $k_{Df} = 1.7 ± 0.2$

m$^2$ g$^{-1}$ as obtained for Dc and Df particles, respectively. These quantities are within and close to the range of values

representative, respectively, for coarse- and fine-dominated dust particles, as reported by the OPAC database

(www.pole-ether.fr): 0.16-0.97 m$^2$ g$^{-1}$ (dust coarse) and 2.3-3.1 m$^2$ g$^{-1}$ (dust fine). Higher MEE values are obtained for

the ND component ($k_{ND} = 3.1 ± 1.3$ m$^2$ g$^{-1}$, in daily average), indicating much smaller particles, and close to that value

of 2.87 m$^2$ g$^{-1}$ reported by OPAC (Hess et al., 1998) for urban aerosols (note that fine polluted aerosols with urban

origin were assumed for the ND component). For comparison, the corresponding mean conversion factors $c_v$ obtained

for Dc and Df particles are, respectively, $c_{v_{Dc}} = 0.8 ± 0.3 \cdot 10^{-12}$ Mm and $c_{v_{Df}} = 0.24 ± 0.02 \cdot 10^{-12}$ Mm, values that are

in good agreement with other reported values (i.e., Mamouri and Ansmann, 2017).

AERONET AOD and AEx values provided along the day (night-time data are assumed equal to the first and last daytime values in each case) also confirm these results (see **Fig. 3a**). In particular, AEx is close to 0.5 (coarse particles

predominance) and higher than 1.5 (fine particles prevalence), respectively, in the first and second part of the day. Regarding LR values as derived from the KF algorithm (**Fig. 3a**, right axis), a daily mean $S_a = 42 \pm 15$ sr is obtained; no significant differences are found between LR values for the first and second part of the day, and just a certain variability is observed along the day as modulated by the dust loading, as expected.

**Figure 4** illustrates, in more detail, both aerosol scenarios before (i.e., at 02:00 UTC) and after (i.e., at 16:00 UTC)

noon (as shown in **Fig. 3** by black arrows), in terms of the profiles of both the particle backscatter coefficients (total $\beta_p$, and $\beta_{Dc}$, $\beta_{Df}$ and $\beta_{ND}$, left panels) and the linear depolarization ratios (volume $\delta^v$ and particle $\delta_p$, right panels). An enhanced dust impact is observed in **Fig. 4a** (02:00 UTC) due to a high amount of Dc particles confined in a layer located between 2 and 5 km height (red line in **Fig. 4a**); contrarily, **Fig. 4b** (16:00 UTC) shows a rather weaker dust incidence from ground up to 4 km height mostly due to a low loading of both Dc and Df particles (red and green lines,

respectively, in **Fig. 4b**), regarded as remains from the passing of the dust intrusion. Indeed, according to HYSPLIT backtrajectories (**Sect. 2.1**), no Saharan origin of air masses is observed for the second part of the day (see **Fig. 1**, d-e panels). AERONET AOD and AEx and KF-derived *LR* values for those different dusty scenarios are also included in **Table 2**. In particular, a $S_a = 50 \pm 10$ sr is retrieved at 02:00 UTC that is within the typical LR range determined for dust, meanwhile a lower value ($S_a = 29 \pm 6$ sr) is found at 16:00 UTC when a rather weaker dust incidence occurs.

Moreover, $\delta_p$ shows values close to the linear particle depolarization ratio for pure Dc particles ($\delta_{Dc}$=0.39) for the first aerosol scenario (**Fig. 4a**, centre panels), and slightly lower than 0.16 ($\delta_{Df}$ for pure dust fine particles) for the second one (**Fig. 4b**, centre panels). In addition, the $\delta_{Df+ND}$ profiles for those times are also shown in **Figure 4** (right panels) in order to examine the corresponding variability of the Df contribution to the particle fine mode with height: $\delta_{Df+ND}$ is greater than 0.10, indicating that the Df fraction within the fine mode is larger than 45.5 %, at altitudes

higher than 1.5 and around 4.0 km height, respectively, for those two dusty situations (**Fig. 4a** and **4b**, respectively), in correspondence with the backscatter profiles; otherwise, Df fraction is reduced (< 40 %) at lower heights. In these two particular cases (**Fig. 4**), the derived MEE values are close to the typical ranges for Dc ($k_{Dc}$: 0.5-0.6) and Df ($k_{Df}$: 1.5-2.0) aerosols (see **Table 3**).

### 3.2 Smoke case

Smoke plumes were observed over BCN station on 23 May 2016. The two principal areas that air masses are arriving from are North America and the Artic, as reported by HYSPLIT backtrajectory analysis for that day at several times (see **Fig. 1**, g-l panels); the smoke origin is likely from forest fires occurred in North America (as stated in **Sect. 2.1**). Hence, the smoke case is examined as a mixture of two components: fine biomass burning particles (SM for smoke) from Canada and USA fires, and another particle type larger than smoke coming from the Arctic region (hereafter,

referred to as non-smoke aerosols, NS). Their vertical separation is achieved using POL-1 retrieval (2-component separation), as described in **Sect. 2.3** and **2.4**. Both the particular backscatter coefficients and mass concentrations are retrieved for each component; in particular, the study is focused only on tropospheric features, avoiding thus aerosols


from other distanced background sources in the boundary layer. Like for the dusty case, **Figure 5** shows the relative fractions of each SM and NS components in terms of the backscatter coefficient and the mass concentration along the

day. Those $k$ values together with the $c_v$ factors at selected times are shown in **Table 3**, as well as the $Pd$ values assumed: 1.30 g m$^{-3}$ for SM and 2.0 g m$^{-3}$ for NS aerosols (see **Sect. 2.4**). Since values of $\delta_p$ higher than 0.1 are found at given altitudes along the day, a high-limit value of the particle linear depolarization ratio for smoke, $\delta_{SM}$, of 0.15 is assumed. This rather high $\delta_{SM}$ value is typical for smoke particles mixed with dust (Tesche et al., 2011; Groβ et al., 2013) as one would expect $\delta_{SM} < 0.10$ for pure biomass burning particles (Müller et al., 2005; Groβ et al., 2013). In

addition, in the first part of the day, AERONET AEx varies between 1.25 and 1.55 (see **Fig. 5a**), indicating rather moderate AEx values as compared to higher fresh smoke values (~ 2.00), as measured by Sicard et al. (2011) also in Barcelona. Hence, the value of $\delta_{SM}$=0.15 reflect a mixing state of biomass burning particles, but not necessarily with dust. For the other, less depolarizing, NS component, a $\delta_{NS}$=0.05 is applied. Those particle linear depolarization ratio values assumed for SM and NS are shown in **Table 2**.

In general, smoke particles are detected during almost all the day, representing approximately 40-60 % of the total height-integrated aerosol backscatter; however, a sharp $\frac{\overline{\beta_{SM}}}{\overline{\beta_p}}$ decrease from those values to around 4 % is observed at 15:00 and 16:00 UTC, also in coincidence with the 47 % decrease found for AEx (see **Fig. 5a**). Since lower AEx values are usually associated to the predominance of large particles and/or to the fine mode decrease, these results are in agreement with that observed reduction of fine biomass burning particles during the same time interval. At those

same times, the $TMC$ reaches high values respect to the daily mean $TMC$ background of $0.05 \pm 0.03$ g m$^{-2}$, that is 0.26 $\pm 0.06$ g m$^{-2}$ in average, as mostly contributed by larger NS aerosols, meanwhile fine SM particles represent only a 3-7 % out of $TMC$ at the same times. In particular, the daily mean $\overline{M_{SM}}$ is $0.017 \pm 0.008$ g m$^{-2}$, representing 2.7 % out of the mean $TMC$ found for the dust case. Regarding KF-derived LR values (see **Fig. 5a**, right axis), a daily mean $S_a$ = $56 \pm 23$ sr is obtained. That value is lower as compared to typical LR of 70 sr for smoke (i.e., Groβ et al., 2013, and

references therein), together with the large relative deviation (42 %) indicates a high aerosol variability along the day, as expected due to the singular arrival of air masses in height and time, and hence the particular vertical aerosol mixing found with the smoke particles.

Regarding the vertical structure, **Figure 6** shows two aerosol scenarios observed along the day: while the smoke appears in clearly defined layers above 5 km height at 06:00 UTC (see **Fig. 6a,** red line), its vertical distribution and

mixing with NS is more heterogeneous at 14:00 UTC (**Fig. 6b**). Indeed, the mean $S_a$ values of $70 \pm 19$ and $35 \pm 9$ sr found, respectively, for the first and second part of the day reflect that the smoke signature detected before noon presents a lower mixing with other aerosols than that observed after noon. Additionally, in average, the mean height-integrated mass concentration for smoke is also obtained in those two different scenarios: $\overline{M_{SM}} = 0.014 \pm 0.002$ and $0.022 \pm 0.009$ g m$^{-2}$ are found, respectively, for the first and second part of the day; those values represent 2.2 and 3.4

%, respectively, out of the $TMC$ found for the intense dust period. In particular, **Figure 6a** clearly shows a smoke layer between 6 and 7.5 km height, also mixed with a certain NS contribution, and presenting $\delta_p$ values of 0.15 and higher. In addition, a smaller SM layer of about 300 m thickness is also found below at around 5.2 km height with rather higher $\delta_p$ than 0.15, and another layer is observed between 3 and 4 km height corresponding to the presence of





NS aerosols with a $\delta_p$ slightly higher than 0.05. The fraction of smoke particles is around 50 % out of total backscatter

(see **Fig. 5a**) with a height-integrated mass concentration for smoke $\overline{M_{SM}} = 0.012 \pm 0.002$ g m⁻², representing 2 % out of the mean $TMC$ during the intense dusty event (see **Table 4**). Later in the day at 14:00 UTC, both SM and NS particles are found along all the profile, being $\delta_p$ values close to 0.15, mainly between 4.0 and 4.5 km height. In addition, a single NS layer is also clearly observed, peaking at 2.5 km height, with $\delta_p$ values decreasing down to 0.05 (see **Fig. 6b**); these results agree with the $\delta_p$ value selected for NS aerosols ($\delta_{NS}$=0.05, see **Table 2**). At this time, a

$\overline{M_{SM}} = 0.023 \pm 0.001$ g m⁻², being 4 % out of the mean $TMC$ for the intense dusty episode, is obtained. Particular LR values for those times shown in **Figure 6** are also included in **Table 2**: $S_a = 81 \pm 16$ sr is retrieved at 06:00 UTC that is within the typical LR range determined for smoke, meanwhile a lower LR ($S_a = 45 \pm 9$ sr) is found at 14:00 UTC, as expected. Besides, particular MEE values derived for smoke particles, $k_{SM} = 4.5 \pm 1.1$ and $1.9 \pm 0.4$ m² g⁻¹ are obtained, respectively, at 06:00 and 14:00 UTC. These results would indicate that smoke plumes detected in the first

scenario are predominantly composed of rather pure fine biomass burning particles with similar MEE values to those reported for Canadian boreal forest fire aged smoke particles (Ichoku and Kaufman, 2005; Reid et al., 2005). However, those observed in the second one would represent a mixed state of smoke particles with an enhanced coarse mode, rather decreasing thus their MEE. All those values are shown in **Tables 3** and **4**.

These results are corroborated by a more detailed analysis of the backtrajectories ending over BCN on 23 May 2016

(selected heights and times of their arrival are shown in **Fig. 1**). In particular, air masses arriving at 06:00 UTC are carrying out smoke particles from Canada and USA fires at altitudes higher than around 4500 m a.s.l. (see **Fig. 1**, h-i panels), while Arctic air masses arrive at lower heights (see **Fig. 1**, g panel). Later on, smoke signature observed at 14:00 UTC is distributed from altitudes higher than around 3000 m a.s.l. height up (**Fig. 1**, k-l panels), and the NS layer identified at around 2500 m height (see **Fig. 6b**) actually corresponds to air masses coming from the Arctic (see

**Fig. 1**, j panel).

### 3.3 Pollen case

The pollination period, i.e., the enhanced formation/presence of pollen particles, in Barcelona is from local sources predominately occurred in March, being the more abundant species such as the *Pinus* and *Platanus* trees (Sicard et al., 2016a). In this case, a pollen episode occurred on 23 March 2016 is selected, corresponding to a high pollination

event observed over BCN (Belmonte, 2016). As for the smoke case, POL-1 retrieval is used to separate pollen (PL) particles from local background (BA) aerosols, mostly composed of urban fine polluted particles. Particle linear depolarization ratios for 'pure' PL, $\delta_{PL}$=0.40, and BA, $\delta_{BA}$=0.05, aerosols are shown in **Table 2**, as well as those $k$ (and $c_v$) values are in **Table 3**. The relative fractions of each aerosol component in terms of the backscatter coefficient and the mass concentration are also calculated along the day.

Pollen signature is clearly observed from 10:00 UTC on, as shown in **Figure 7** by the increase of their relative fraction $\frac{\overline{\beta_{PL}}}{\overline{\beta_p}}$, with a maximum around 30 % between 12:00 and 16:00 UTC. The coincident increase of AEx (see **Fig. 7a**) is probably associated to the formation of local urban aerosols, which are much smaller particles as compared to pollen grains. This hypothesis suggests that local urban aerosols dominate the columnar-averaged optical properties.



Regarding the LR, a mean value of $S_a$ = 55 ± 17 sr is obtained during the pollen occurrence, while $S_a$ = 71 ± 17 sr is
found for the no pollen detection period. That $S_a$ value for pollen is close to that considered in other works (Sicard et
al., 2016a). The fraction of the height-integrated mass concentration for pollen $\overline{M_{PL}}$ respect to the $TMC$ reaches a
maximum of around 40 % at 15:00 UTC; in addition, the $TMC$ evolution is fairly constant with a daily-averaged $TMC$
of 0.029 ± 0.003 g m⁻², being the mean $\overline{M_{PL}}$ = 0.007 ± 0.003 g m⁻², i.e., 25 % out of $TMC$, in the 12:00-23:00 UTC
interval. For comparison, these $TMC$ levels represent only 1.1 % of the dust $TMC$ during their higher dust incidence,
as discussed in **Sect. 3.1**. Regarding the MEE derived for pollen particles, a mean $k_{PL}$ = 2.4 ± 0.8 m² g⁻¹ is obtained.
Sicard et al. (2016a) estimated a $k_{PL}$ = 3.2 m² g⁻¹ considering an effective radius size of 24 μm for the pollen grains
registered during a pollination episode in March 2015 (data not shown). Hence, the $k_{PL}$ value found in this work can
be in agreement with that estimated value as long as pollen particles detected in our case are larger than those observed
by Sicard et al (2016a), as MEE decreases as particle size increases.

The vertical distribution at two particular times along the day is shown in **Figure 8**. No pollen particles are
significantly detected at 10:00 UTC (**Fig. 8a**), only local aerosols, with low $\delta_p$ values close to 0.05 from surface up to
around 1 km height, slightly increasing from that altitude up, likely due to uplofted particles. The pollen presence is
clearly observed at 15:00 UTC (**Fig. 8b**): $\delta_p$ increases, reaching higher values between 0.10 and 0.15, and pollen
particles are mostly confined up to 1.5 km height from the surface. The corresponding mass loading for pollen $\overline{M_{PL}}$
at this time is 0.011 ± 0.003 g m⁻² (see **Table 4)**.

**4 Conclusions**

The synergetic use of the POLIPHON retrieval with P-MPL measurements is introduced for the first time in order to
separate dust (both coarse Dc, and fine Df, modes) and biomass burning smoke (SM) particles from their mixtures
with other aerosols (namely, non-dust ND, and non-smoke NS aerosols); in addition, a case study of pollen (PL)
detection detached from local urban background aerosols (BA) is also examined. In all the cases, the particle linear
depolarization ratio for each aerosol 'pure' component is a relevant constraint by using POLIPHON method: the
separation of aerosol mixtures into their particle components can be performed just for rather different depolarising
particles. In particular, typical linear depolarization ratios found in the literature are assumed for each pure aerosol
component: 0.39, 0.16 and 0.05, respectively, for Dc, Df and ND; 0.15 and 0.05, respectively, for SM and NS; and
0.40 and 0.05, respectively, for PL and BA.

In this work, a good performance is achieved by obtaining the relative optical and mass contributions of each aerosol
component along the day as based on P-MPL continuous 24/7 observations carried out in Barcelona (NE Spain): three
case studies observed on 5 July, 23 May and 23 March 2016 are examined, respectively, for dust, smoke and pollen
occurrences. In particular, the POLIPHON 1-step version (POL-1: separation into two components) is applied for the
smoke and pollen cases; in order to illustrate the 3-component separation for the dust case, a combined algorithm
using both the POLIPHON 1-step (POL-1) and 2-step (POL-2) versions (namely POL-1/2) is described in more detail.
In addition, both the vertical and columnar particle depolarization ratio for the total fine (Df+ND) mode, $\delta_{Df+N}$ , and





correspondingly both the vertical and columnar fraction of Df particles to the total fine (Df+ND) mode, are also
estimated by using the POL-1/2 retrieval (the a priori assumption of those variables is thus avoided). Indeed, minimal

differences in the particle backscatter coefficient $\beta$ for each dusty and non-dusty component are found as obtained
from either POL-1 or POL-1/2 approaches, as long as a vertical depolarization ratio for the total fine (Df+ND) mode
$\delta_{Df+}$ (z) is regarded; otherwise, the use of a single columnar, no height-resolved, $\delta_{Df+}^c$ is inadequate due to the
plausible Df variability, respect to the total fine mode, with height.

Moreover, the extinction-to-mass conversion procedure is described in terms of the Mass Extinction Efficiency (MEE:

$k$, m$^2$ g$^{-1}$), a parameter associated to the size of the particles. The MEE is estimated for each aerosol component by
using the corresponding conversion factors as calculated from AERONET data (volume concentrations and
extinctions for the coarse and fine modes), as reported at simultaneous times with P-MPL measurements, and the
particles densities assumed for each type of aerosol. In addition, the effective MEE ($k_{eff}$, a measure of the
predominant size of those aerosol mixtures) is also retrieved for each aerosol event. Hence, height-integrated mass

concentrations (i.e., mass loadings, g m$^{-2}$) are obtained along the day for each component. In general, the daily
evolution of their relative optical and mass contributions, respect to the height-integrated total backscatter coefficient
and total mass concentration (total mass loading) for each aerosol case is also derived. Due to the variation of the
aerosol situation observed for each case study along the day, particular different aerosol scenarios can be present, and
hence their vertical distribution are examined in more detail in this work.

In the dust case, a Saharan dust intrusion arrives at BCN during the first part of the day, meanwhile a weak dust
incidence is observed for the second part of the day, as also confirmed by AERONET data and HYSPLIT
backtrajectory analysis. This is due to the predominance of large particles (Dc component) during the first half of the
day. In terms of mean dust mass loading, values of $TMC = 0.6 \pm 0.1$ and $0.2 \pm 0.1$ g m$^{-2}$ are obtained, respectively, at
time intervals before and after noon: this last value just represents a mass loading of 34 % respect to that found for the

first part of the day. In addition, mean MEE values of $k_{Dc} = 0.5 \pm 0.1$ m$^2$ g$^{-1}$ and $k_{Dc} = 1.7 \pm 0.2$ m$^2$ g$^{-1}$ are obtained
for Dc and Df particles, respectively. These quantities are within and close to the range of values representative of
coarse- and fine-dominated dust particles, respectively. AERONET AOD and AEx values reported along the day
confirm these results; in particular, AEx is close to 0.5 (coarse particles predominance) and higher than 1.5 (fine
particles prevalence), respectively, in the first and second part of the day. A mean KF-derived lidar ratio $S_a = 42 \pm 15$

sr is obtained with no significant differences for the first and second part of the day. Regarding particular aerosol
scenarios, a $S_a = 50 \pm 10$ sr is retrieved at 02:00 UTC (within the typical range of lidar ratios defined for dust),
meanwhile a lower value ($S_a = 29 \pm 6$ sr) is found at 16:00 UTC when a rather weaker dust incidence occurs. Moreover,
$\delta_p$ shows values close to the particle linear depolarization ratio for pure Dc particles (0.39) for the first dusty scenario,
and lower than 0.16 (typical for pure dust fine particles), highlighting the prevalence of ND aerosols, for the second

one. In addition, the particle depolarization ratio for the total fine (Df+ND) mode is greater than 0.10, that is, the
relative Df fraction within the total fine mode is larger than 45.5 %, at altitudes higher than 1.5 and around 4.0 km
height, respectively, for those two particular dusty situations. The derived MEE values are typical for Dc ($k_{Dc}$: 0.5-
0.6) and Df ($k_{Dc}$: 1.5-2.0) aerosols in those two particular cases.



In the smoke case, the air masses arriving over BCN on 23 May 2016 are mainly coming from two areas: North
America and the Artic, as reported by HYSPLIT backtrajectory analysis. Hence, fine biomass burning particles
originated from fires occurred in Canada and USA are likely mixed with other larger than smoke aerosols coming
from the Arctic region (non-smoke aerosols, NS). In general, both SM and NS particles are found along all the profile;
$\delta_p$ values are higher than 0.10 and close to 0.15 when SM particles are mostly detected. Fine smoke particles are
observed during almost all the day, representing approximately 40-60 % of the total height-integrated aerosol
backscatter coefficient; the mean mass loading for smoke is $\overline{M_{SM}} = 0.017 \pm 0.008$ g m$^{-2}$, representing 2.7 % out of that
mean $TMC$ found for the dust case. However, individual decreases in the relative smoke fractions of both the
backscatter coefficient and mass concentration are also observed along the day, coinciding also in time with AEx
decreases (as associated to predominance/reduction of coarse/fine particles). Regarding the vertical structure, two
aerosol scenarios are observed along the day: the smoke signature is specially detected at defined layers in the
beginning of the daytime, while a vertical SM distribution mixed along with a NS layered structure is observed later
on. Mean LR values of $S_a = 70 \pm 19$ and $35 \pm 9$ sr are found, respectively, for the first and second part of the day,
showing a lower smoke mixing before than after noon. In addition, the mean mass loading for smoke as obtained in
those two different scenarios is $\overline{M_{SM}} = 0.014 \pm 0.002$ and $0.022 \pm 0.009$ g m$^{-2}$, respectively, for the first and second
part of the day, i.e., 2.2 and 3.4 %, respectively, out of the $TMC$ found for the intense dust period. This is likely due
to the singular arrival of air masses in height and time, and hence the particular vertical aerosol mixing found together
with the smoke particles over BCN. Besides, the corresponding particular MEE values derived for smoke particles in
those two scenarios are $k_{SM} = 4.5 \pm 1.1$ and $1.9 \pm 0.4$ m$^2$ g$^{-1}$, respectively, indicating that smoke plumes detected in
the first scenario are predominantly composed of rather pure fine biomass burning particles, unlike the situation in the
second one with a mixed state of smoke particles with an enhanced coarse mode.

In the pollen case, the PL signature is clearly observed from 10:00 UTC on, when the relative fraction of the height-
integrated backscatter coefficient for pollen enhances, reaching a maximum around 30 % between 12:00 and 16:00
UTC, and $\delta_p$ increases with values between 0.10 and 0.15 from the surface up to around 1.5 km height. A mean LR
of $S_a = 55 \pm 17$ sr is obtained during the pollen occurrence period; this value is close to that considered by other
authors. The relative fraction of mass loading for pollen reaches a maximum of around 40 % at 15:00 UTC, being the
mass loading of $\overline{M_{PL}} = 0.011 \pm 0.003$ g m$^{-2}$, i.e., 1.7 % out of that for dust during their higher incidence at that time.
In addition, the mean MEE derived for pollen particles is $k_{PL} = 2.4 \pm 0.8$ m$^2$ g$^{-1}$, representing an intermediate value
between those reported for Df particles ($k_{Df} = 1.7 \pm 0.2$ m$^2$ g$^{-1}$) and for smaller local background urban polluted
aerosols ($k_{BA} = 3.4 \pm 0.7$ m$^2$ g$^{-1}$). However, the $k_{PL}$ can reach higher/lower values depending on a prevalent
smaller/larger size of the pollen grains.

In summary, the vertical separation of aerosol mixtures into their components is achieved by using the POLIPHON
retrieval in synergy with continuous 24/7 P-MPL measurements. The methodology, including the extinction-to-mass
conversion procedure, is described and applied to several aerosol mixtures case studies. Therefore, vertical optical and
mass features are obtained in a daily basis for different climate-relevant aerosols: dust, smoke and pollen particles. In
addition, the method can be relatively easily applicable to spaceborne lidars with an equivalent configuration (elastic



with a depolarization-sensitive channel) such as the ongoing CALIOP/CALIPSO, and the forthcoming ATLID/EarthCARE (future ESA mission to be launched in 2019).

**Appendix A. List of acronyms.**

| Symbol (*) (**) | Parameter | Units |
|---|---|---|
| $ch_{co}$, $ch_{cross}$ | P-MPL signal channels: co-polar and cross-polar, respectively | a.u. |
| $P^{tot}$, $P^p$, $P^s$ | P-MPL range-corrected signals: total, parallel, perpendicular signals, respectively ($P^{tot} = P^p + P^s = ch_{co} + 2\,ch_{cross}$) | a.u. |
| $\beta_p$ | Total particle backscatter coefficient | km$^{-1}$ sr$^{-1}$ |
| $\beta_i$ | Backscatter coefficient for a specific particle component ($i$) | km$^{-1}$ sr$^{-1}$ |
| $\overline{\beta_p}$ | Height-integrated total particle backscatter coefficient | sr$^{-1}$ |
| $\overline{\beta_i}$ | Height integrated backscatter coefficient for a specific particle component ($i$) | sr$^{-1}$ |
| $\beta_{mol}$ | Molecular backscatter coefficient | km$^{-1}$ sr$^{-1}$ |
| $\Delta$ | Root square differences (see Eq. 5) | km$^{-1}$ sr$^{-1}$ |
| $\tilde{\Delta}$ | Root mean square differences (see Eq. 7) | sr$^{-1}$ |
| $\delta^V$ | Linear volume depolarization ratio | --- |
| $\delta_p$ | Linear particle depolarization ratio | --- |
| $\delta_i$ | Linear particle depolarization ratio for a specific particle component ($i$) | --- |
| $\delta_{mol}$ | Molecular depolarization ratio | --- |
| $\delta_{Df+ND}$ | Total fine (Df+ND) depolarization ratio (residual depolarization ratio) | --- |
| $\delta^c_{Df+ND}$ | Columnar total fine (Df+ND) depolarization ratio | --- |
| $R$ | Backscattering ratio ($= \frac{\beta_{mol}+\beta_p}{\beta_{mol}}$) | --- |
| $S_a$ | Lidar Ratio (LR) (KF-derived) | sr |
| $\sigma_p$ | Total particle extinction coefficient | km$^{-1}$ |
| $\sigma_i$ | Extinction coefficient for a specific particle component ($i$) | km$^{-1}$ |
| AOD | Aerosol Optical Depth (total particle extinction, AERONET data) | --- |
| AEx | Angstrom Exponent (AERONET data) | --- |
| $k_{eff}$ | Effective Mass Extinction Efficiency (MEE) | m$^2$ g$^{-1}$ |
| $k_i$ | Mass Extinction Efficiency for a specific particle component ($i$) | m$^2$ g$^{-1}$ |
| $c_{v_x}$ | Extinction-to-volume conversion factor for a specific particle size mode | $10^{-12}$ Mm |
| $VC_x$ | Volume concentration for a specific particle size mode (AERONET data) | $10^{-12}$ Mm |
| $\tau_x$ | Extinction for a specific particle size mode (AERONET data) | --- |
| $TMC$ | Total Mass Concentration | g m$^{-3}$ |



| $M_i$ | Mass concentration for a specific particle component ($i$) | g m$^{-3}$ |
| $\overline{TMC}$ | Total mass loading (height-integrated $TMC$, over-bar is removed for simplicity) | g m$^{-2}$ |
| $\overline{M_i}$ | Mass loading (height-integrated $M_i$) for a specific particle component ($i$) | g m$^{-2}$ |

(*) $i$ denotes the aerosol component: dust coarse (Dc), dust fine (Df), non-dust (ND), smoke (SM), non-smoke (NS),
pollen (PL), background aerosols (BA).
(**) $x$ denotes the particle size mode: coarse (c), fine (f).

**Acknowledgements**

This work is supported by the Spanish Ministerio de Economía y Competitividad (MINECO) under grant CGL2014-
55230-R (AVATAR project) and the ACTRIS-2 (Aerosols, Clouds, and Trace Gases Research Infrastructure
Network) Research Infrastructure Project funded by the European Union's Horizon 2020 research and innovation
programme (grant agreement n. 654109). Lidar measurements in Barcelona were also supported by the Spanish
MINECO (project TEC2015-63832-P) and EFRD (European Fund for Regional Development); by the Department of
Economy and Knowledge of the Catalan autonomous government (grant 2014 SGR 583); and the Unidad de
Excelencia Maria de Maeztu (project MDM-2016-0600) financed by the Spanish Agencia Estatal de Investigación.
The authors gratefully acknowledge the NOAA Air Resources Laboratory (ARL) for the provision of the HYSPLIT
transport and dispersion model and/or READY website (http://www.ready.noaa.gov) used in this publication. C. C.-
J. thanks the Ministerio de Educación, Cultura y Deporte (MECD) support under grant PRX15/00375 for the 3-month
research stay at TROPOS (Germany); and A. del A. thanks the MINECO support (Programa de Ayudas a la Promoción
del Empleo Joven e Implantación de la Garantía Juvenil en i+D+i) under grant PEJ-2014-A-52129.

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






**Table 1. Relative uncertainties for the P-MPL-derived particle optical properties (at 532 nm wavelength), and mass concentrations. (n) and (d) stand for night-time and day-time P-MPL measurements, respectively.**

| Parameter | Symbol (*) | Relative uncertainty (%) | References |
|---|---|---|---|
| Particle backscatter coefficient (km$^{-1}$ sr$^{-1}$) | $\beta_p$ | 5 - 20 (n), 10 - 30 (d) | Rocadenbosch et al. (2012) |
| Particle extinction coefficient (km$^{-1}$) | $\sigma_p$ | 10 – 30 (n), 15 – 40 (d) | Derived from the errors in $\beta_p$ and $LR$ |
| Lidar ratio (sr) | $LR$ | 5 - 10 | Derived from KF algorithm |
| Particle linear depolarization ratio | $\delta_p$ | 10 - 60 | Rodríguez-Gómez et al. (2017) |
| Volume linear depolarization ratio | $\delta^V$ | 10 - 50 | Derived from the errors in both $pRCS$ and $sRCS$ |
| Total Mass Concentration (g m$^{-3}$) | $TMC$ | 10 - 40 | Derived from the error in AOD ($=\sum_z \sigma_p(z)$), mainly |

(*) As denoted in the text.



**Table 2. Aerosol cases observed over BCN on selected days. AERONET data at particular times of the event (as shown in Figs. 4, 6 and 8), including those KF-retrieved LR values ($S_a$), and parameters used in the POLIPHON retrieval algorithm, depending on the version applied. References for the assumed particle linear depolarization ratio for specific components are also included $\delta_i$ (either $i$ = 1-3, or $i$ = 1, 2, depending on the case) are also included. Errors are shown in parenthesis.**

| Aerosol case Date | Time (UTC) | $S_a$ (sr) | AERONET data | | POLIPHON retrieval (*) | Linear depolarization ratio for each aerosol component (**) | | | |
|---|---|---|---|---|---|---|---|---|---|
| | | | AOD | AEx | | $\delta_1$ | $\delta_2$ | $\delta_3$ | Reference |
| DUST 05 July 2016 | 02:00 | 50 (10) | 0.33 (0.01) | 0.52 (0.03) | POL-1 | 0.31 (DD) | 0.05 (ND) | --- | Tesche et al. (2011); Ansmann et al. (2012) |
| | 16:00 | 29 (6) | 0.25 (0.01) | 1.70 (0.01) | POL-2 | 0.39 (Dc) | 0.16 (Df) | 0.05 (ND) | Mamouri and Ansmann (2014) |
| SMOKE 23 May 2016 | 06:00 | 81 (16) | 0.14 (0.02) | 1.30 (0.24) | POL-1 | 0.15 (SM) | 0.05 (BA) | --- | Groβ et al. (2013) |
| | 14:00 | 45 (9) | 0.16 (0.01) | 0.72 (0.05) | | | | | |
| POLLEN 23 March 2016 | 10:00 | 98 (20) | 0.12 (0.01) | 0.75 (0.02) | POL-1 | 0.40 (PL) | 0.05 (BA) | --- | Sicard et al. (2016) |
| | 15:00 | 39 (8) | 0.10 (0.01) | 1.74 (0.03) | | | | | |

(*) POL-1: Separation of two components; POL-2: Separation of three components.

(**) Particular $\delta_i$ values assumed for each specific aerosol component ($i$), regarded as 'pure' aerosols: Dc, Df and ND stand, respectively, for dust coarse, dust fine and non-dust particles; SM and NS stand, respectively, for smoke and non-smoke aerosols; and PL and BA stand, respectively, for pollen particles and local background aerosols.




**Table 3. Parameters involved in the extinction-to-mass conversion for each aerosol case: the AERONET-reported and derived mass conversion factors ($c_v$), the assumed particle densities ($Pd$), and the Mass Extinction Efficiency ($k$) values. For the dust case (3-component separation): $i$ = 1 (Dc), 2 (Df) and 3 (ND); and for the smoke / pollen cases (2-component separation), respectively: $i$ = 1 (SM / PL) and 2 (NS / BA). Errors are shown in parenthesis.**


| Aerosol case | Time (UTC) | AERONET data (*) | | $c_v$ ($10^{-12}$ Mm) | | | $Pd$ (g cm$^{-3}$) | $k$ (m$^2$ g$^{-1}$) | | | |
|---|---|---|---|---|---|---|---|---|---|---|---|
| | | $VC_c$ $VC_f$ | $\tau_c$ $\tau_f$ | 1 | 2 | 3 | | 1 | 2 | 3 | eff |
| DUST (POL-1/2) | 02:00 | 0.192 (0.003) 0.022 (0.009) | 0.237 (0.006) 0.100 (0.003) | 0.81 (0.03) | 0.20 (0.09) | --- (---) | 2.60 (Dc, Df) 1.80 (ND) | 0.47 (0.02) | 2.0 (0.9) | --- (---) | 0.57 (0.07) |
| | 16:00 | 0.062 (0.003) 0.040 (0.003) | 0.092 (0.003) 0.181 (0.001) | 0.67 (0.05) | 0.25 (0.02) | 0.20 (0.01) | | 0.57 (0.05) | 1.5 (0.1) | 2.7 (0.1) | 1.6 (0.2) |
| SMOKE (POL-1) | 06:00 | 0.005 (0.001) 0.021 (0.006) | 0.024 (0.001) 0.122 (0.002) | 0.17 (0.05) | 0.21 (0.05) | --- | 1.30 (SM) 2.00 (NS) | 4.5 (1.4) | 2.4 (0.5) | --- | 3.5 (1.5) |
| | 14:00 | 0.049 (0.001) 0.027 (0.006) | 0.062 (0.001) 0.066 (0.001) | 0.41 (0.10) | 0.79 (0.03) | --- | | 1.9 (0.5) | 0.63 (0.02) | --- | 2.1 (0.4) |
| POLLEN (POL-1) | 10:00 | 0.013 (0.002) 0.012 (0.002) | 0.058 (0.010) 0.054 (0.001) | 0.22 (0.07) | 0.22 (0.04) | --- | 0.92 (PL) (*Platanus*) 1.80 (BA) | 4.9 (1.6) | 2.5 (0.5) | --- | 4.1 (1.2) |
| | 15:00 | 0.017 (0.001) 0.012 (0.001) | 0.035 (0.001) 0.070 (0.004) | 0.47 (0.03) | 0.17 (0.02) | --- | | 2.3 (0.1) | 3.2 (0.5) | --- | 3.5 (1.0) |

(*) 'c' and 'f' denote the particle coarse and fine modes, respectively.




**Table 4. Height-integrated mass concentration ($\overline{M}_t$, i.e., mass loading, g m$^{-2}$) for each component and the total mass concentration ($TMC$) at two times for each aerosol case. Errors are shown in parenthesis.**


| Aerosol case | Time (UTC) | $\overline{M}$ (g m$^{-2}$) | | | $TMC$ (g m$^{-2}$) |
|---|---|---|---|---|---|
| | | 1 | 2 | 3 | |
| DUST | 02:00 | 0.54 (0.04) | 0.03 (0.02) | --- (---) | 0.57 (0.05) |
| | 16:00 | 0.08 (0.01) | 0.026 (0.003) | 0.057 (0.003) | 0.16 (0.02) |
| SMOKE | 06:00 | 0.012 (0.004) | 0.027 (0.007) | --- | 0.04 (0.01) |
| | 14:00 | 0.023 (0.006) | 0.053 (0.004) | --- | 0.08 (0.01) |
| POLLEN | 10:00 | 0.0009 (0.0003) | 0.029 (0.006) | --- | 0.029 (0.006) |
| | 15:00 | 0.011 (0.001) | 0.017 (0.004) | --- | 0.028 (0.005) |





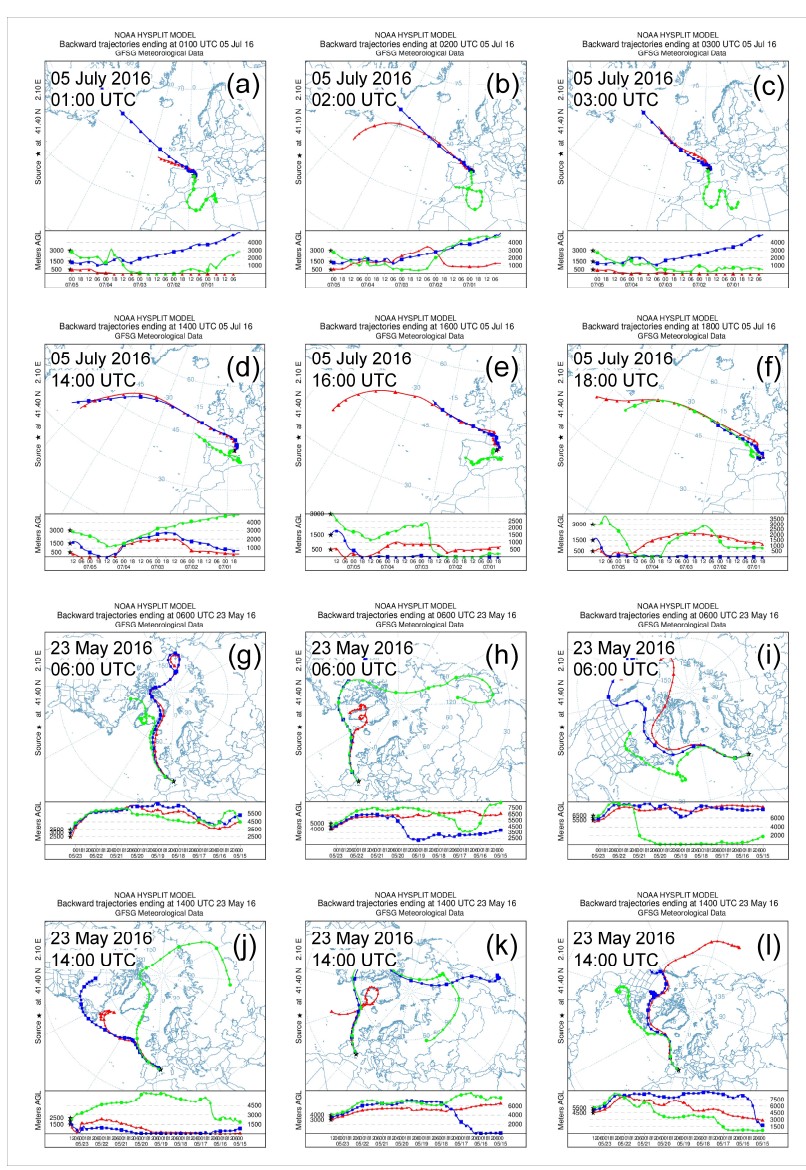


**Figure 1: HYSPLIT backtrajectories ending at different altitudes over BCN depending on the aerosol case (only for the dust and smoke cases): (a) – (f) for dust (5 days back) on 5 July 2016; (g) - (l) for smoke (10 days back) on 23 May 2016. Selected times of the air masses arrivals are related to those aerosol profiles particularly examined (as shown in Sect. 3; in particular, see Figs. 4, 6 and 8).**






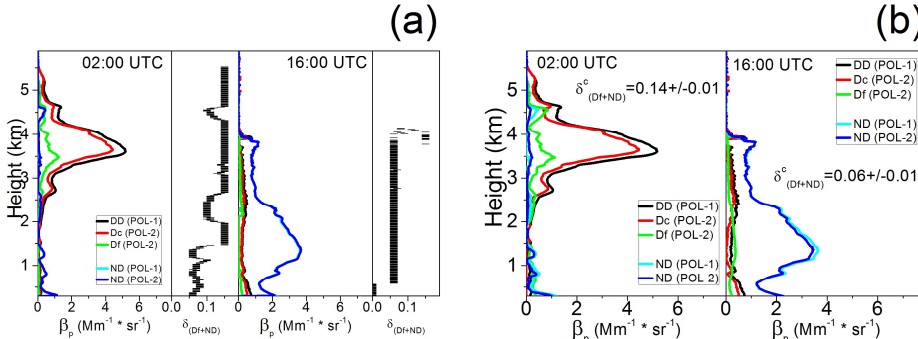

**Figure 2. POL-1 versus POL-2 differences in particle backscatter coefficient profiles for each component (total dust $\beta_{DD}$, and non-dusty $\beta_{ND}$ from POL-1; dust coarse $\beta_{Dc}$ and fine $\beta_{Df}$, being $\beta_{Dc} + \beta_{Df} = \beta_{DD}$, and non-dusty $\beta_{ND}$ from POL-2) retrieved for the dust case on 5 July 2016 at 02:00 and 16:00 UTC, respectively, by using (optimally-derived): (a) a $\delta_{Df+ND}(z)$.profile, and (b) a single columnar $\delta_{Df+N}^{c}$  value.**






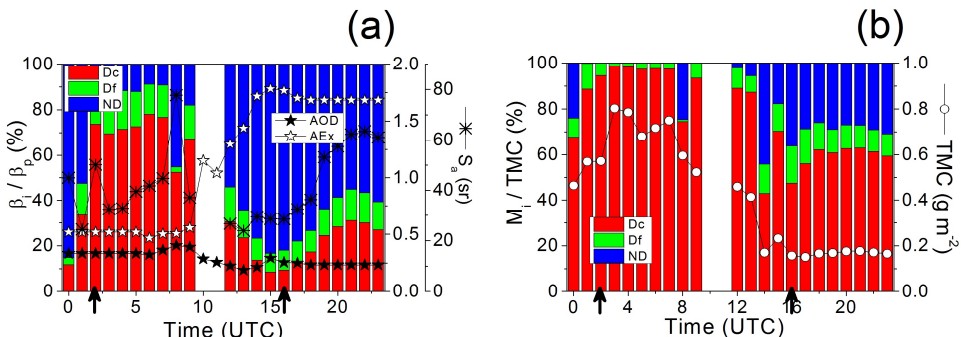

**Figure 3. Dust event occurred on 5 July 2016. Evolution of the relative contribution (a) $\frac{\overline{\beta_i}}{\overline{\beta_p}}$ (%) and (b) $\overline{M_i}/TMC$ (%) (the bar over the variable are removed in the figure for clarity) for each aerosol component along the day: Dc (red bars), Df (green bars) and ND (blue bars) which denote, respectively, dust coarse, dust fine and non-dusty aerosols. In plot (a) (right axis) AERONET hourly-averaged AOD and AEx (black and white stars, respectively) and KF-derived $S_a$ (lidar ratio, sr; cross symbols) values are reported; in plot (b) (right axis) $TMC$ (total mass loading, g m^{-2}; open circles) is also included.**

**Black arrows on the time axis indicate selected times for which vertical profiles are shown in Fig. 4.**






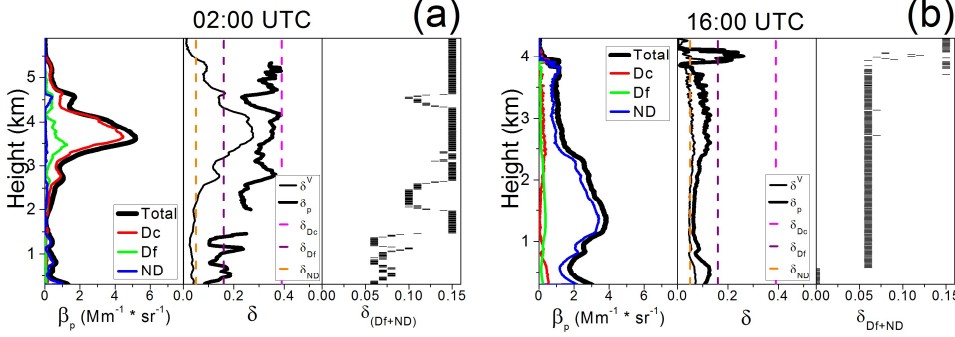

**Figure 4. Dust event occurred on 5 July 2016. Vertical profiles of the particle backscatter coefficients (total and for each specific component; left panels), the linear depolarization ratios (volume $\delta^V$ and particle $\delta_p$; centre panels), and the estimated depolarization ratio for the fine (Df+ND) mode ($\delta_{Df+ND}$, right panels) at two times illustrating the different aerosol scenario observed along the day: (a) at 02:00 UTC (high dust incidence), and (b) at 16:00 UTC (low dust incidence). Specific depolarization ratios selected for each pure aerosol component are also shown by vertical dashed lines (see legend) in the centre panels.**




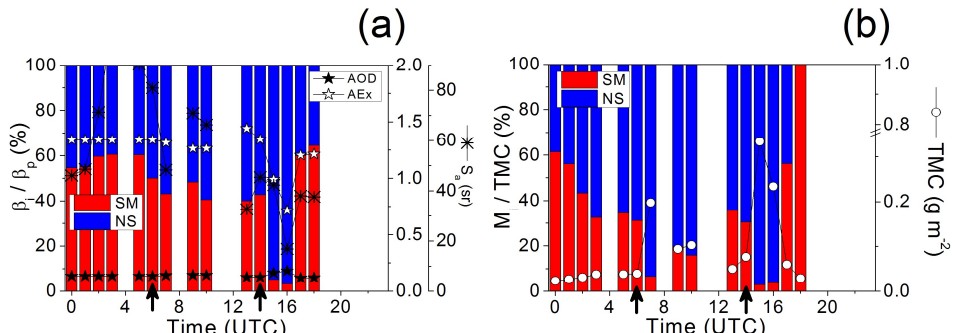

**Figure 5. The same as Fig. 3, but for the smoke case occurred on 23 May 2016: SM (red bars) and NS (blue bars), which**
**denote, respectively, smoke and non-smoke components. Black arrows on the time axis indicate selected times for which**
**vertical profiles are shown in Fig. 6.**






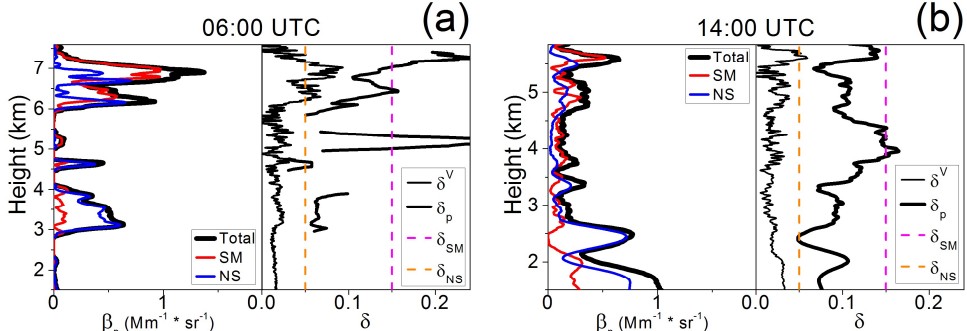

Figure 6. The same as Fig. 4, but for the smoke event occurred on 23 May 2016 at: (a) 06:00 UTC, and (b) 14:00 UTC. Specific depolarization ratios selected for each smoke aerosol component are also shown by vertical dashed lines (see legend for details).






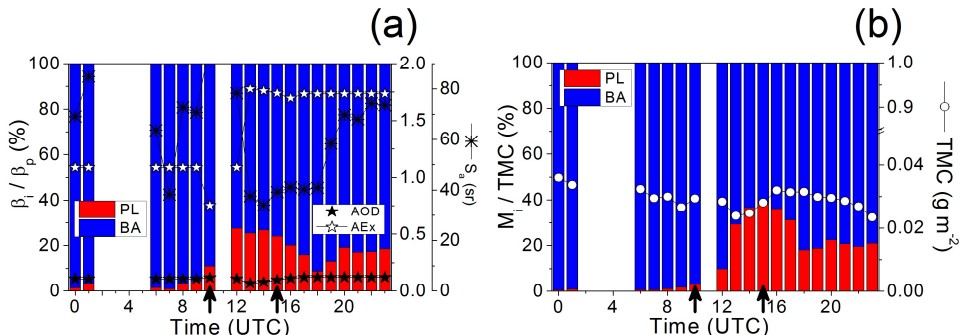

**Figure 7. The same as Fig. 3, but for the pollen event occurred on 23 March 2016: PL (red bars) and BA (blue bars), which denote, respectively, pollen and local background aerosol components. Black arrows on the time axis indicate selected times for which vertical profiles are shown in Fig. 8.**





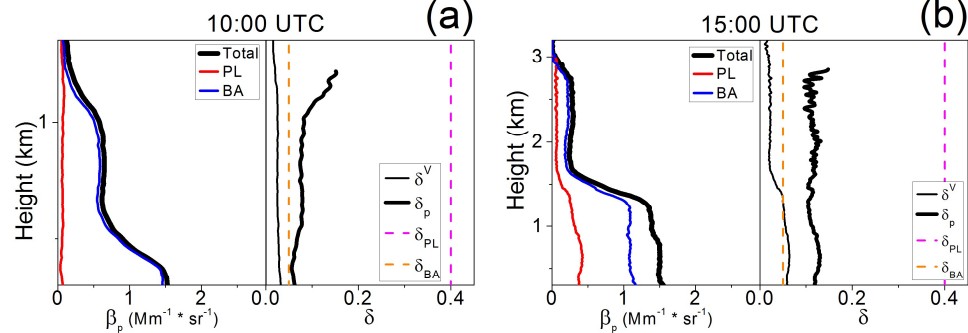

**Figure 8. The same as Fig. 4, but for the pollen event occurred on 23 March 2016 at: (a) 10:00 UTC (no PL detection), and (b) 15:00 UTC (enhanced PL occurrence). Specific depolarization ratios selected for each pure aerosol component are also**
**shown by vertical dashed lines (see legend for details).**