# Peer review of "Separation of the optical and mass features of particle components in different aerosol mixtures by using POLIPHON retrievals in synergy with continuous polarized Micro-Pulse Lidar (P-MPL) measurements"

_Atmospheric Measurement Techniques, 2018_

## Referee Comment (RC1) · Anonymous Referee #3 · 24 May 2018

The work is of great scientific significance applicable across many fields (e.g. air chemistry, radiative transfer, health & human impact), and research method is sound. However, technical & grammatical errors were wide spread making the manuscript, at times, difficult to read. If not for these I would suggest minor revisions, but spelling, verb-tense, and phrasing errors are abound warranting major revisions. -Further justification needs to be made for some parts. Why are the given times for the examples profiles chosen? The maximum and minimum AOD for each case along with their respective times is discussed at length, but these aren't necessarily the times in the profiles shown in figures 4, 6, and 8. - Referring to diurnal variations as "First" and "second" part of the day is poor wording, and is relative. During discussion of figures 3, 5, and 7 stated times are sometimes unclear if you mean local time or UTC with wording such as "noon," which is a relative term to local time. It is also difficult to see the black AOD and Lidar Ratio symbols on the dark blue color bar in these figures. Figures 3a and 3b don't have matching x-axes. -Why was the smoke case broken up into smoke and non-smoke, and the pollen case broken up into pollen and background aerosol? The non-smoke aerosol is said to be of arctic origin, but there is no mention of potential local background aerosol in the retrieval, isn't this a possibility? Vice versa for the pollen case, why is there no HYSPLIT analysis for the pollen case? Is it assumed on this day the background aerosol didn't have an origin outside BCN? All this needs justification.

---

## Referee Comment (RC2) · Anonymous Referee #4 · 2 Jun 2018

Cordoba-Jabonero et al. P-MPL Aerosol Discrimination

This paper describes adaptation of the POLIPHON technique to polarized micro pulse lidar instruments with the goal of continuous 24/7 monitoring and discrimination of various aerosol types and their physical and optical properties based primarily on the depolarization ratio. There is obvious merit to this work. MPLs are the workhorse of the ground-based lidar community. Algorithms based on a technique like this could

prove invaluable to long-term data records, both for climatological analysis and satellite verification. The paper is very well organized and the figures clear and legible. This is the first time this reviewer has seen this manuscript.

My summary recommendation is that this paper be accepted pending Minor Revisions. I have only a few scientific concerns. Technically, there are issues with the language. I've tried to help (see attachment). In particular, pay attention to the use of colons and semi-colons. Most of the usage is redundant and/or inappropriate. Also, be mindful of paragraph structure, as it is very important in ensuring a logical and consistent flow for your reader!

Scientifically...a few points:

- Are you bound by corresponding AERONET inversion retrievals, and thus a minimum AOD of $\sim 0.40$ in order to conduct your retrievals? If so, what is the impact of this?

- What at the prospects for adapting this technique operationally? This is never really discussed.

- To my belief, and in spite of some papers in the literature, your definition of volume depolarization is not correct in the classical sense. Follow Sassen (1991) for reasoning and historical evolution of the term. This doesn't matter, really, but you should be clear what it is that you've defined.

A very good paper and study. Congratulations.

Please also note the supplement to this comment:
https://www.atmos-meas-tech-discuss.net/amt-2018-15/amt-2018-15-RC2-supplement.pdf

**Supplement:**

[revised manuscript text omitted]

---

## Author Comment (AC1) · 12 Jun 2018

"*Separation of the optical and mass features of particle components in different aerosol mixtures by using POLIPHON retrievals in synergy with continuous polarized Micro-Pulse Lidar (P-MPL) measurements*" by Carmen Córdoba-Jabonero et al., Atmos. Meas. Tech. Discuss., doi:10.5194/amt-2018-15, 2018.

**Authors' response (in blue) to the Reviewer#3's comments (in italic black):**

Authors thank the comments and suggestions of the reviewer that definitely will improve the manuscript (see revised version).

Next, the authors' response to the specific referee's comments is addressed.

*The work is of great scientific significance applicable across many fields (e.g. air chemistry, radiative transfer, health & human impact), and research method is sound. However, technical & grammatical errors were wide spread making the manuscript, at times, difficult to read. If not for these I would suggest minor revisions, but spelling, verb-tense, and phrasing errors are abound warranting major revisions.*

A complete revision has been performed, and the modifications have been implemented using the Word 'Track Changes' tool in the manuscript. Furthermore, Copernicus copy-editing will certainly improve language issues in the final publication process.

Figures 1, 3, 5 and 7 have been also modified in order to introduce the changes suggested by the referee.

*- Further justification needs to be made for some parts. Why are the given times for the examples profiles chosen? The maximum and minimum AOD for each case along with their respective times is discussed at length, but these aren't necessarily the times in the profiles shown in figures 4, 6, and 8.*

The purpose to select those specific times is to show two different atmospheric situations in terms of the vertical distribution of the optical properties (backscatter coefficients and depolarization ratios) of the aerosols for each case (dust, smoke and pollen, as shown in Figures 4, 6 and 8, respectively), and not specified by the maximal and minimal AOD values only. Therefore, for more clarity, the text has been modified (using the Word 'Track Changes' tool) in some parts of the manuscript. That is:

The text in the first version of the manuscript in **page 13, lines 414-416** has been replaced by the following one:

"In order to illustrate the vertical distribution of dust particles, **Figure 4** shows an example in terms of the profiles of both the particle backscatter coefficients (total $\beta_p$, and $\beta_{Dc}$, $\beta_{Df}$ and $\beta_{ND}$, left panels) and the linear depolarization ratios (volume $\delta^v$ and particle $\delta_p$, right panels) of both aerosol scenarios: 1) when the dust event presents a high incidence as occurred for instance at 02:00 UTC (**Fig. 4a**); and 2) after the dust particles are almost

completely removed (i.e., situation observed at 16:00 UTC, see **Fig. 4b**). These scenarios are also indicated in **Figure 3** by black arrows."

The text in the first version of the manuscript in **page 14, lines 468-470** has been replaced by the following one:

"Regarding the vertical structure, **Figure 6** shows examples of two different aerosol scenarios observed on the day: 1) a well-defined smoke layer is observed, for instance, between 6 and 7.5 km height with a certain mixing with NS aerosols at 06:00 UTC (see **Fig. 6a,** red line); and 2) the smoke signature can be detected highly mixed with NS aerosols along the atmospheric profile (i.e., situation observed at 14:00 UTC, see **Fig. 6b**). These both scenarios are also indicated in **Figure 5** by black arrows."

The text in the first version of the manuscript in **page 16, lines 525-529** has been replaced by the following one:

"In order to display the vertical distribution for this case, profiles of the particle backscatter coefficients and both the volume and particle linear depolarization ratios are shown in **Figure 8** (see legend inside). For instance, the vertical distribution is shown at 10:00 UTC, when no pollen particles are significantly detected (**Fig. 8a**), with low $\delta_p$ values close to 0.05 from surface up to around 1 km height and slightly increasing from that altitude up. This is likely due to uplifted particles. In comparison, the situation occurred later on the day (i.e., that observed at 15:00 UTC, **Fig. 8b**), the amount of pollen clearly enhances: $\delta_p$ increases, reaching higher values between 0.10 and 0.15, and pollen particles are mostly confined up to 1.5 km height from the surface. These two scenarios are also indicated in **Figure 7** by black arrows."

*- Referring to diurnal variations as "First" and "second" part of the day is poor wording, and is relative. During discussion of figures 3, 5, and 7 stated times are sometimes unclear if you mean local time or UTC with wording such as "noon," which is a relative term to local time. It is also difficult to see the black AOD and Lidar Ratio symbols on the dark blue color bar in these figures. Figures 3a and 3b don't have matching x-axes.*

Authors thank the suggestions of the reviewer. Hence, the manuscript has been changed (using the Word 'Track Changes' tool) as follows:

- The text has been modified regarding the revision performed for the wording of 'first/second part of the day' and 'noon' terms.
- Symbols denoting the AOD and Lidar Ratio in Figures 3, 5, and 7 have been replaced for more clarity.
- Figure 3 has been modified to match the x-axes in panels a and b.

*- Why was the smoke case broken up into smoke and non-smoke, and the pollen case broken up into pollen and background aerosol? The non-smoke aerosol is said to be of*

*arctic origin, but there is no mention of potential local background aerosol in the retrieval, isn't this a possibility? Vice versa for the pollen case, why is there no HYSPLIT analysis for the pollen case? Is it assumed on this day the background aerosol didn't have an origin outside BCN? All this needs justification.*

The arrival of smoke plumes over BCN is mostly at altitudes above the boundary layer (BL), as stated in the manuscript. Hence, the 'smoke' study was examined as a two-component case: smoke and non-smoke aerosols (likely from Arctic origin, as analyzed in the manuscript), but focused only on those tropospheric features above the boundary layer (BL), thus disregarding aerosols from other plausible local background BL sources (also stated in the manuscript). The pollen case is slightly different. The pollen particles are originating from local pollination events usually occurring close to the surface. Thus the pollen is likely mixed with aerosols supposedly coming from background, local sources. These background (BA) aerosols are supposed to be mostly composed of urban fine polluted particles, and their exact origin, whether they are local or not, is not relevant, since they do not depolarize and cannot be mistaken for highly depolarizing pollen particles.

For clarifying this aspect, the text has been modified (using the Word 'Track Changes' tool) in some parts of the manuscript, that is:

The text in the first version of the manuscript in **pages 13-14, lines 441-443** has been replaced by the following one:

"Both the particular backscatter coefficients and mass concentrations are retrieved for each component. In particular, the arrival of smoke plumes over BCN is mostly at altitudes above the boundary layer (BL); hence, this case is focused only on those tropospheric features above the BL, thus disregarding aerosols from other plausible local background BL sources."

The text in the first version of the manuscript in **page 15, lines 505-506** has been replaced by the following one:

"As for the smoke case, POL-1 retrieval is used to separate pollen (PL) particles from background (BA) aerosols. These BA are supposed to be mostly composed of urban fine polluted particles, and their exact origin, whether they are local or not, is not relevant since they do not depolarize and cannot be mistaken for highly depolarizing pollen particles. This is also the reason why HYSPLIT backtrajectories were not calculated."

---

## Author Comment (AC2) · 12 Jun 2018

*Separation of the optical and mass features of particle components in different aerosol mixtures by using POLIPHON retrievals in synergy with continuous polarized Micro-Pulse Lidar (P-MPL) measurements*" by Carmen Córdoba-Jabonero et al., Atmos. Meas. Tech. Discuss., doi:10.5194/amt-2018-15, 2018.

**Authors' response (in blue) to the Reviewer#4's comments (in italic black):**

*My summary recommendation is that this paper be accepted pending Minor Revisions. I have only a few scientific concerns. Technically, there are issues with the language. I've tried to help (see attachment). In particular, pay attention to the use of colons and semi-colons. Most of the usage is redundant and/or inappropriate. Also, be mindful of paragraph structure, as it is very important in ensuring a logical and consistent flow for your reader!*

Authors are grateful for the comments and suggestions of the reviewer, and mainly, for the changes related to the language as proposed in the supplement. All that will definitely improve the manuscript (see revised version).

A complete revision has been performed, and the modifications have been implemented using the Word 'Track Changes' tool in the manuscript. Nevertheless, we are sure that Copernicus copy-editing will furthermore improve language.

Figures 1, 3, 5 and 7 have been also modified in order to introduce the changes suggested by the referee.

Next, the authors' response to the specific referee's comments is addressed.

*- Are you bound by corresponding AERONET inversion retrievals, and thus a minimum AOD of ~0.40 in order to conduct your retrievals? If so, what is the impact of this?*

An AOD ~ 0.40 is only obtained for the dust case (intense dust scenario). For the weak dust period and the other two cases (smoke and pollen), the AOD is lower than usually during such events (disregarding single strong smoke episodes). The parameters needed for the retrieval (in general, the extinction-to-mass conversion factors) were those provided by AERONET, if available, independently on the AOD derived. We used AERONET V2 inversion Level 1.5 data for all our aerosol cases; hence, the AOD ~ 0.4 threshold limitation does not apply. We selected this AERONET data, in particular, due to the unavailability of the almucantar-derived data from V3 inversion at any level and those scarce data from V2 at Level 2.0.

Then, the text in the first version of the manuscript in **page 6, lines 197-198** has been replaced by the following one:

"AERONET V2 inversion Level 1.5 data were used for all the aerosol cases due to the unavailability of the almucantar-derived data from V3 inversion at any level and those scarce data from V2 at Level 2.0. Hence, the threshold limitation of AOD > 0.4 does not apply. Both AOD and the Ångström exponent (AEx) together with other AERONET parameters used in this work were also hourly-averaged in order to coincide with the 1-h averaging applied to P-MPL measurements."

*- What at the prospects for adapting this technique operationally? This is never really discussed.*

Actually, the procedure can be easily applicable in other MPLs operating within the extended MPLNET network, but also adapted to the space-borne lidars. In order to introduce this point, the following text in the **Abstract** in the first version of the manuscript (**page 2, lines 45-47**) has been added:

"In fact, this procedure can be simply implemented in other P-MPLs also operating within the world-wide Micro-Pulse Lidar Network (MPLNET), thus extending the aerosol discrimination at a global scale."

And also the text in the first version of the manuscript in **pages 18-19, lines 618-621** has been replaced by the following one:

"It should be noted that the method can be relatively easily applicable to other P-MPLs also within the world-wide NASA/Micro-Pulse Lidar Network (MPLNET), since all those systems present the same instrumental and operating configuration. Hence, the aerosol discrimination can be extended at a global scale. In addition, it can be also adapted to spaceborne lidars with an equivalent configuration (elastic with a depolarization-sensitive channel) such as the ongoing CALIOP/CALIPSO, and the forthcoming ATLID/EarthCARE (future ESA mission to be launched in 2019)."

*- To my belief, and in spite of some papers in the literature, your definition of volume depolarization is not correct in the classical sense. Follow Sassen (1991) for reasoning*

*and historical evolution of the term. This doesn't matter, really, but you should be clear what it is that you've defined.*

The **section 2.2 ('Polarized Micro-Pulse lidar (P-MPL) system')** has been modified in order to clarify this point. In particular, the **text** in the first version of the manuscript **in page 5, lines 164-177** has been replaced by the following one:

"Polarization capabilities rely on the collection of two-channel measurements (i.e., the signal measured in the so-called relative 'co-polar' and 'cross-polar' channels of the instrument, denoted as $P_{co}(z)$ and $P_{cr}(z)$ signals, respectively; see Sigma Space Corp. Manual, 2012, for more details). By adapting the methodology described in Flynn et al. (2007), the parallel and perpendicular P-MPL range-corrected signals (RCS, also called Normalized-Relative-Backscatter signals, $NRB$), represented as $P^{||}(z)$ and $P^{\perp}(z)$, respectively, can be expressed in terms of those P-MPL co- and cross-channel signals, $P_{co}(z)$ and $P_{cr}(z)$, respectively, as (hereafter, the dependence with height is omitted for simplicity)

$$P^{||} = P_{co} + P_{cr}, \tag{1}$$

and

$$P^{\perp} = P_{cr} \tag{2}$$

Then, the total RCS, $P$, can be expressed as

$$P = P^{||} + P^{\perp} = P_{co} + 2\,P_{cr}. \tag{3}$$

Final corrected $P$, $P^{||}$ and $P^{\perp}$ are obtained using the procedure described in Campbell et al. (2002) and Welton and Campbell (2002). The linear volume depolarization ratio, $\delta^V$, in a classical sense (Sassen, 1991), can be defined as

$$\delta^V = \frac{P^{\perp}}{P^{||}}. \tag{4}$$

Then, the linear volume depolarization ratio $\delta^V$ for a MPL system (Flynn et al., 2007) can be easily expressed as

$$\delta^V = \frac{P_{cr}}{P_{co}+P_{cr}}. \tag{5}$$"

These changes have affected the order of the Equations, then they have been re-numbered (also in the text).